# A quantitative method of resolving annual precipitation for the past millennia from Tibetan ice cores

**Wangbin Zhang[1], Shugui Hou[1,2,3], Shuang-Ye Wu[4], Hongxi Pang[1,2], Sharon B. Sneed[5], Elena V. Korotkikh[5], Paul A. Mayewski[5], Theo M. Jenk[6,7], and Margit Schwikowski[6,7]**

[1]School of Geography and Ocean Science, Nanjing University, Nanjing 210023, China

[2]Collaborative Innovation Center of Climate Change, Jiangsu Province, Nanjing, China

[3]School of Oceanography, Shanghai Jiao Tong University, Shanghai 200240, China

[4]Department of Geology and Environmental Geosciences, University of Dayton, Dayton, OH 45469, USA

[5]Climate Change Institute, University of Maine, Orono, ME 04469, USA

[6]Laboratory of Environmental Chemistry, Paul Scherrer Institute, 5232 Villigen PSI, Switzerland

[7]Oeschger Centre for Climate Change Research, University of Bern, Sidlerstrasse 5, 3012 Bern, Switzerland

**Correspondence:** Shugui Hou (shugui@nju.edu.cn, shuguihou@sjtu.edu.cn)

**Abstract.** Net accumulation records derived from alpine ice cores provide the most direct measurement of past precipitation. However, quantitative reconstruction of accumulation for past millennia remains challenging due to the difficulty in identifying annual layers in the deeper sections of ice cores. In this study, we propose a quantitative method to reconstruct annual accumulation from alpine ice cores for past millennia, using as an example an ice core drilled at the Chongce ice cap in the northwestern Tibetan Plateau (TP). First, we used laser ablation inductively coupled plasma mass spectrometry (LA-ICP-MS) technology to develop ultra-high-resolution trace element records in three sections of the ice core and identified annual layers in each section based on seasonality of these elements. Second, based on nine [14]C ages determined for this ice core, we applied a two-parameter flow model to established the thinning parameter of this ice core. Finally, we converted the thickness of annual layers in the three sample sections to past accumulation rates based on the thinning parameter derived from the ice flow model. Our results show that the mean annual accumulation rates for the three sample sections are $109 \, \mathrm{mm \, yr^{-1}}$ (2511–2541 years BP), $74 \, \mathrm{mm \, yr^{-1}}$ (1682–1697 years BP), and $68 \, \mathrm{mm \, yr^{-1}}$ (781–789 years BP), respectively. For comparison, the Holocene mean precipitation is $103 \, \mathrm{mm \, yr^{-1}}$. This method has the potential to reconstruct continuous high-resolution precipitation records covering millennia or even longer time periods.

## 1 Introduction

Precipitation, including both snowfall and rainfall, is a crucial component of the Earth's energy and water cycles and is a variable parameter associated with atmospheric circulation in weather and climate studies (Kidd and Huffman, 2011). Accurate and reliable knowledge of precipitation is of paramount importance not only for the study of water resource management, but also for understanding and monitoring the Earth's climate change (Kidd and Huffman, 2011; Sun et al., 2018). The earliest systematic instrumental observations of precipitation (i.e., rain gauges) began in the 18th century in Europe, but they were not put in place until much later in other parts of the world (Sun et al., 2018). Our knowledge of precipitation in earlier times therefore relies on precipitation-sensitive proxy records from different types of biological and geological archives, e.g., tree rings, stalagmite, terrestrial and marine sediment, and ice cores (Cai et al., 2012; Kaspari et al., 2008; Thompson et al., 1995; Xu et al., 2019; Yang et al., 2014).

Net accumulation recorded in alpine ice cores provides the most direct measurement for past precipitation, as glaciers are formed by accumulating annual layers of snow (Paterson and Waddington, 1984). However, in order to derive accurate net accumulation records, sampling resolution must be high enough to obtain reliable layer thickness information for the relevant timescale (typically annual). The most com-

mon approach is to obtain annual-layer thickness based on the seasonal cycles of ice core parameters, such as the stable isotope ratio of oxygen in the water ($\delta^{18}O$), the concentration of major ions (e.g., $Ca^{2+}$, $Mg^{2+}$, $NH_4^+$, and $SO_4^{2-}$), and the presence of melt layers (Thompson et al., 2018). In addition, the nonlinear thinning of annual layers caused by ice flow must be suitably compensated (Bolzan, 1985; Henderson et al., 2006; Roberts et al., 2015). This thinning parameter of ice cores is usually derived from an ice flow model constrained by the ages of absolute chronological markers, e.g., peak of beta and/or tritium activity from thermonuclear bomb testing in the second half of the 20th century, well-defined aerosol layers and/or tephra from large volcanic eruptions, and radioactive dating method based on $^{210}Pb$ activity decay (Uglietti et al., 2016; Zhang et al., 2015). Using these conventional methods, a great number of accumulation records were developed from alpine ice cores over the past decades, covering decades to centuries (e.g., Hardy et al., 2003; Henderson et al., 2006; Hou et al., 2002; Kaspari et al., 2008; Yao et al., 2008). However, it remains challenging to develop annually resolved accumulation records covering longer (e.g., millennial) time periods because of the difficulties in identifying annual layers and obtaining accurate chronologies in the deeper part of alpine ice cores due to rapid thinning (e.g., Henderson et al., 2006; Yao et al., 2008). During the past 2 decades, several effective methods, for example, continuous flow analysis (CFA) technology and laser ablation inductively coupled plasma mass spectrometry (LA-ICP-MS) technology, were developed to measure the chemicals preserved in ice cores with millimeter to sub-millimeter sampling resolution. The resulting ultra-high-resolution records could resolve seasonal signals of chemical constituents in ice cores and were increasingly used to accurately discern annual layers of ice cores from Antarctica, Greenland, and the Alps (e.g., Bohleber et al., 2018; Clifford et al., 2019; Haines et al., 2016; Massam et al., 2017; More et al., 2017; Winstrup et al., 2019). The remaining challenge for reconstructing long-term accumulation records thus lies in establishing accurate thinning parameters, and this is largely dependent on the reliable dating of alpine ice cores, particularly at deeper sections. Recently, a novel method was developed to extract water-insoluble organic carbon (WIOC) particles at microgram level from carbonaceous aerosol embedded in the glacier ice for accelerator mass spectrometry (AMS) $^{14}C$ dating (Uglietti et al., 2016). Carbonaceous aerosols are constantly transported to high-altitude glaciers, where they are deposited and eventually incorporated into the glacier ice. Consequently, carbonaceous aerosols in ice cores can provide reliable dating at any given depth when the samples contain sufficient WIOC ($> 10\,\mu g$). These dates can then be used to constrain an ice flow model to estimate the mean accumulation rate and thinning of alpine ice cores.

In this study, we propose a quantitative method to establish annual accumulation records of past millennia, taking three sections from an ice core drilled at the Chongce ice cap in the northwestern Tibetan Plateau (TP) as an example (Fig. 1). First, we measured the concentration of aluminum (Al), calcium (Ca), iron (Fe), sodium (Na), magnesium (Mg), copper (Cu), and lead (Pb) from three sections of the Chongce ice core using LA-ICP-MS technology. It is worth noting that this is the first application of ultra-high-resolution LA-ICP-MS measurement for Tibetan ice cores. Based on the seasonal cycles of these elements, we identified annual layers in the ice core and measured their thickness. Second, we derived the thinning parameter and the mean accumulation rate of the entire Chongce ice core using a two-parameter steady-state flow model constrained by the $^{14}C$ ages and the ages of other reference layers (e.g., $\beta$ activity peak). Based on the results, we calculated the modeled annual layer thickness for mean accumulation at different depths. Finally, we derived the actual accumulation for each annual layer within the three sample sections as the product of the ratio of the observed to modeled annual layer thickness and the average annual accumulation of this ice core. It is worth noting that this is also the first time that annual layer identification based on seasonal cycles of LA-ICP-MS intensity of multiple elements is combined with annual layer thinning modeling to reconstruct annual precipitation records at the millennial timescale from the Tibetan ice cores. In addition, we evaluated the reliability of this method by comparing our results with other reconstructed and modeled precipitation series for the TP.

## 2 Materials and methods

### 2.1 The Chongce ice cores

The Chongce ice cap is located in the western Kunlun Mountains on the northwestern TP (Fig. 1), covering an area of $163.06\,km^2$ with a volume of $38.16\,km^3$ (Shi, 2008). The ice cap faces south, with a mean equilibrium line altitude of 5900 m above sea level (a.s.l.) (Fig. 1). Climate of the Chongce ice cap and its vicinity is largely controlled by the strength of the mid-tropospheric westerlies (Pang et al., 2020) (Fig. S1 in the Supplement). Based on the High Asia Refined analysis (HAR) data (available at https://www.klima.tu-berlin.de/, last access: 27 December 2021) (Maussion et al., 2014), precipitation over the Chongce ice cap is highly seasonal (Figs. S2 and S3). Summertime precipitation accounts for $\sim 28\,\%$ of the annual total, whereas the amount from December to May accounts for $\sim 59\,\%$ (Figs. S3 and S4). Autumn (September to November) has the lowest amount (13 %) of precipitation.

In October 2012, we retrieved two ice cores to bedrock with lengths of 134.03 m (Core 1; 35°14′5.77″ N, 81°7′15.34″ E; 6010 m a.s.l.) and 135.81 m (Core 2; 35°14′6.11″ N, 81°6′50.62″ E; 6010 m a.s.l.) and a shallow ice core with a length of 58.82 m (Core 3; 35°14′5.69″ N, 81°6′51.71″ E; 6010 m a.s.l.) from the Chongce ice cap with an electromechanical drill (Hou et al., 2018, 2019). The

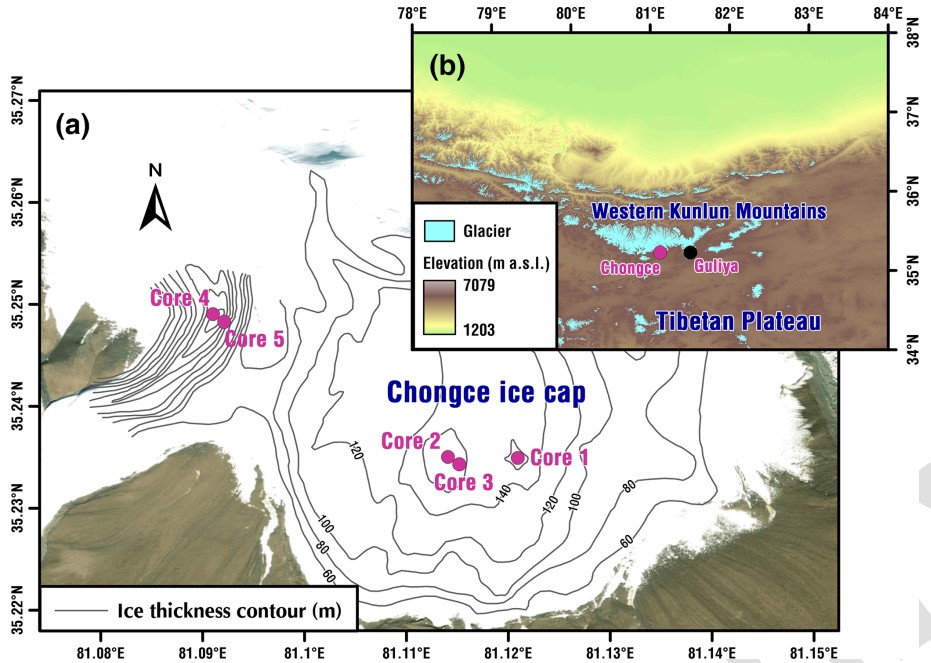

**Figure 1. (a)** Satellite image of the Chongce ice cap with location of the ice core drilling sites (deep rose dots). Ice thickness contours were superimposed on the image of the glacier. **(b)** Topographic map of the northwestern TP. The satellite imagery map is available at https://www.mapsofworld.com/satellite-map/world.html (last access: 13 July 2018). The topographic data were extracted using the Shuttle Radar Topography Mission (SRTM) 90 m DEM digital elevation database, available at http://srtm.csi.cgiar.org/ (last access: 13 July 2018). Data of glaciers are from the Global Land Ice Measurements from Space (GLIMS; available at http://www.glims.org, last access: 13 July 2018, GLIMS and NSIDC, 2015).

distance between the drilling sites of Core 2 and Core 3 is ∼ 2 m. In October 2013, two more ice cores to bedrock were recovered from the Guozha glacier on the same ice cap with lengths of 216.61 m (Core 4; 35°14′56.58″ N, 81°5′27.70″ E; 6105 m a.s.l.) and 208.63 m (Core 5; 35°14′56.00″ N, 81°5′28.06″ E; 6104 m a.s.l.) (Fig. 1). Borehole temperatures are −12.8, −12.6 and −12.6° at 10 m depth for Core 1, Core 2, and Core 3, and −8.8 and −8.8° at 130 m depth for Core 1 and Core 2, respectively (Fig. S4), suggesting that the Chongce ice cap is frozen to the bedrock (Hou et al., 2018). The density profiles of Core 2, Core 3, and Core 4 are shown in Fig. S5. All the ice cores were kept frozen and transported to the cold laboratory (−20°C) at Nanjing University for further processing.

## 2.2 The LA-ICP-MS analysis

The LA-ICP-MS analysis was conducted at the W. M. Keck Laser Ice Core Facility of the Climate Change Institute, University of Maine, following procedures presented by Sneed et al. (2015) and Spaulding et al. (2017). The system includes the following components: a Thermo Element 2 ICP-MS instrument (Thermo Fisher Scientific, Bremen, Germany), coupled to a standard New Wave UP-213 laser ablation system (New Wave Research, Fremont, California, USA), and the Sayre Cell™, a cryocell chamber designed to hold 1 m of ice

at a temperature of −15 °C. The chamber is equipped with a small (∼ 5 cm$^3$) open-design ablation chamber for continuous ablation of ice core samples at close to their original length (Spaulding et al., 2017).

We selected three trial sections from Core 2 (Section I, II, and III) for the ultra-high-resolution LA-ICP-MS analysis, as well as the top section of Core 3 (Top Section) for conventional chemical analysis (∼ 3 cm sampling resolution) for comparison (An et al., 2016). The depths of these sections are 0–10.030 m for the Top Section (Core 3), 72.645–73.151 m for Section I, 107.977–108.389 m for Section II, and 122.670–123.032 m for Section III (Core 2) (An et al., 2016). Prior to analysis, Section I, II, and III were split axially into two halves using a band saw. One half was stored as an archive. For the remaining half, we first scraped the surface (longitudinal section) with a Lie-Nielsen stainless-steel blade to remove possible contamination caused by previous sample preparation. The ice sample was then placed in the W. M. Keck Laser Ice Facility Sayre Cell™ whilst the system was purged with argon (Ar) gas for 2 min to remove impurities in the system. To save analysis time, the multi-element method of LA-ICP-MS analysis (Spaulding et al., 2017) was used to measure Na, Mg, Cu, and Pb from an ablated line along the surface of the ice core and Al, Ca, and Fe from a parallel line. These two ablated lines were separated by 200 µm to prevent any possible overlap. The sampling res-

olution is 153 µm per sample. The unit of LA-ICP-MS measurements is intensity (counts per second, cps).

## 2.3 The $\beta$ activity measurements

A total of 31 samples were collected successively from the top to a depth of 14.805 m of the Chongce Core 3 (Table S1). Each sample is about 1 kg. The $\beta$ activity was measured using an Alpha-Beta Multidetector (Mini 20, Eurisys Mesures) at the State Key Laboratory of Cryospheric Science, Lanzhou, China. Details about $\beta$ activity measurements can be found in An et al. (2016).

## 2.4 The $^{14}$C measurements

The $^{14}$C measurements were made at the Paul Scherrer Institute and the University of Bern (LARA laboratory), Switzerland. From the Chongce Core 2, nine samples were selected for $^{14}$C dating (Hou et al., 2018) using a method based on $^{14}$C determination in the water-insoluble organic carbon fraction (WIOC) of the aerosols in the ice (Sigl et al., 2009). In brief, we first decontaminated the ice for the $^{14}$C samples by removing the $\sim 3$ mm outer layer using a stainless-steel bandsaw in a cold room ($-20$°C) and rinsing the remaining ice core samples with ultrapure water (MilliQ, 18.2 MΩ cm quality) in a laminar flow hood. The samples were then melted to collect the water-insoluble carbonaceous particles contained in the ice by filtration. The filters were subsequently combusted at 340°C and then 650°C to separate organic carbon (OC) from element carbon (EC). The resulting $CO_2$ was measured by the Mini Carbon Dating System (MICADAS) with a gas ion source for $^{14}$C analysis. Details about sample preparation and WIOC separation can be found in Uglietti et al. (2016). The overall procedural blanks were estimated using artificial ice blocks of frozen ultrapure water, which were treated the same way as real ice samples. The average overall procedural blank is $1.34 \pm 0.62$ µg carbon with a F$^{14}$C of $0.69 \pm 0.13$ (Uglietti et al., 2016). The conventional $^{14}$C ages were calibrated using the OxCal v4.4 online program (https://c14.arch.ox.ac.uk/, last access: 24 November 2021) (Ramsey and Lee, 2013) with the IntCal13 calibration curve (Reimer et al., 2013). Results of the individual samples can be found in Table S2.

## 2.5 Annual-layer identification using the StratiCounter program

In comparison to our visual annual-layer results, we applied the StratiCounter program to identify annual layers for Section III (see Sect. 3.1). The StratiCounter program is an automated annual-layer detection method based on the hidden Markov model (HMM) algorithms (Winstrup et al., 2012). The code for the StratiCounter program is available at the GitHub repository (http://www.github.com/maiwinstrup/StratiCounter, last access: 14 September 2020; Winstrup, 2015; Winstrup et al., 2012).

## 2.6 The TraCE-21ka simulation

For comparison, we also used data from the "Simulation of Transient Climate Evolution over the Last 21 000 years" (TraCE-21ka) (Collins et al., 2006; Liu et al., 2009). The TraCE-21ka experiment was performed using a coupled ocean–atmosphere model, the Community Climate Model version3 (CCSM3), forced by realistic variations in insolation, atmospheric greenhouse gases (GHGs), meltwater fluxes, and continental ice sheets. The atmospheric resolution is $3.75° \times 3.75°$ horizontally, with 26 vertical levels. we calculated annual precipitation in the western Kunlun Mountains covering the last 3 millennia based on outputs from the TraCE-21ka climate simulation (available at https://www.earthsystemgrid.org, last access: 26 July 2019).

# 3 Results

## 3.1 Annual-layer identification using multiple chemical species

Various chemical species obtained from Tibetan ice cores exhibit distinct seasonal cycles (An et al., 2016; Thompson et al., 2018). On the northwestern TP, the $\delta^{18}$O values in modern precipitation show distinct seasonal fluctuations, with high values in summer and low values in winter (Thompson et al., 2018). In addition, chemical elements (e.g., Al, Ca, Fe, and Mg) also show marked seasonal cycles, with high concentrations in late winter and spring and low concentrations in summer (Thompson et al., 2018).

In this paper, annual layers of the Top Section were identified based on seasonal cycles of $\delta^{18}$O values (Fig. 2). The distance between two adjacent low $\delta^{18}$O values was defined as the annual layer thickness (An et al., 2016). The result was verified by a reference of $\beta$ activity peak in 1963 CE due to thermonuclear bomb testing and a second $\beta$ activity peak in 1986 CE corresponding to the Chernobyl nuclear accident. The derived average annual layer thickness of the Top Section is $168.61 \pm 61.91$ mm (corresponding to 140.76 mm w.e.). Annual layers of Section I, II, and III were identified based on seasonal cycles of Al, Ca, Fe, and Mg. In these sections, the LA-ICP-MS profile of each element (Al, Ca, Fe, and Mg) is characterized by the regular occurrence of several distinct peaks grouped together, along with an elevated baseline of each element's concentration (Fig. 2). The groups of peaks are separated by a prolonged section of low element concentrations (Fig. 2). These grouped peaks are interpreted as independent snow events with elevated element concentrations or with wind-blown dust deposition between these snow events (Fig. 2). The periods of low values correspond to snow deposition during the summer (Fig. 2). Therefore, annual layer boundaries were defined as synchronous local maxima (maximum in each group of peaks) in all four element concentrations. In addition, we follow

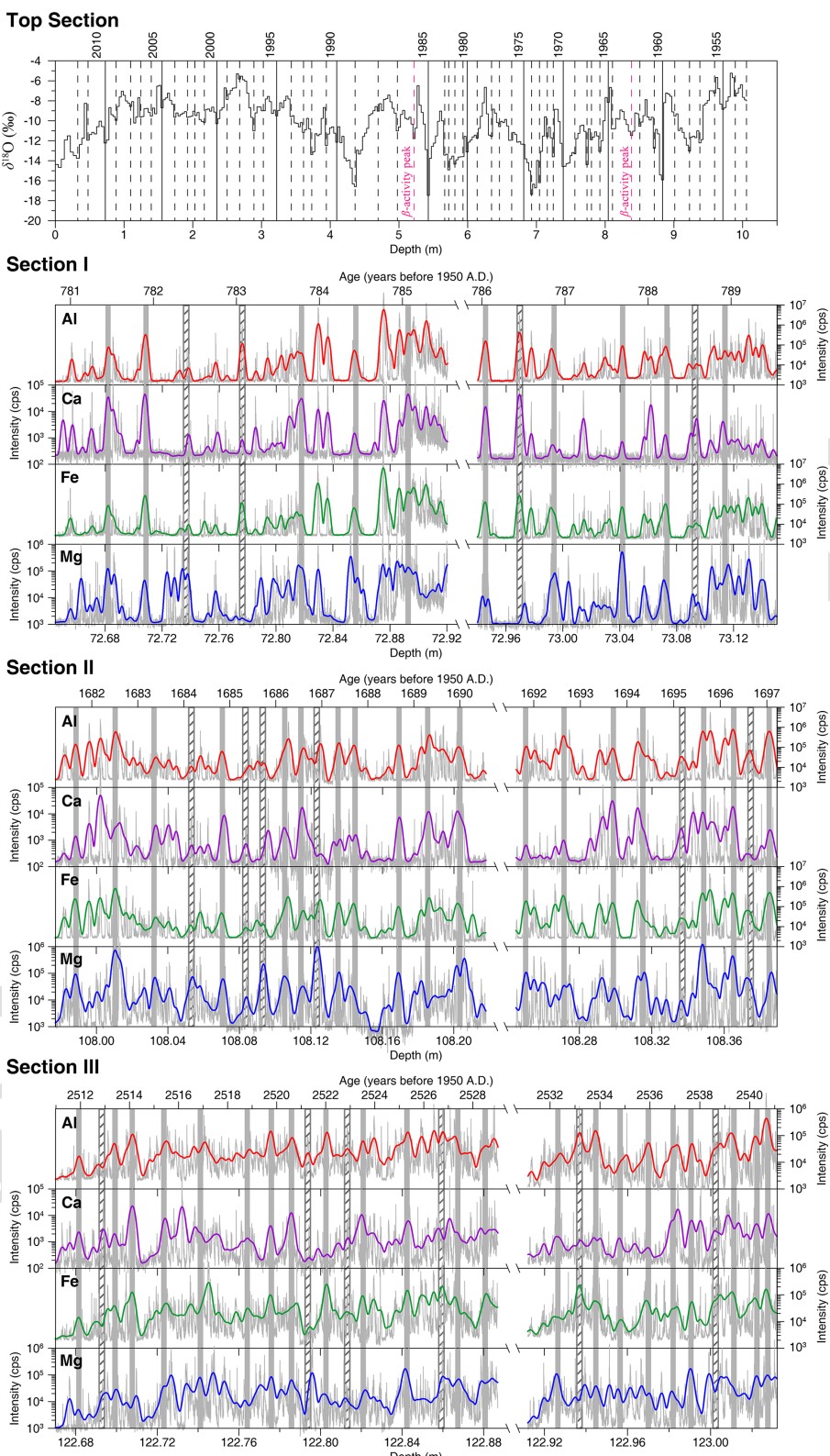

**Figure 2.** Annual layer counting for the Top Section and Section I, II, and III (top to bottom). The annual layers of the Top Section are identified based on the seasonality of $\delta^{18}$O and two $\beta$ activity peaks (An et al., 2016). The annual layers of Section I, II, and III are marked at the winter and spring peaks (grey bars) of Al, Ca, Fe, and Mg concentrations. The open grey bars filled with forward slashes indicate uncertain annual layers. Thin grey lines indicate raw data, and thick colored lines represent 200-point Gaussian smoothing. LA-ICP-MS intensity is reported in counts per second.

the approach successfully employed for Greenland ice cores by Rasmussen et al. (2006) to quantify counting uncertainty from uncertain layers. In this approach, we count uncertain layers as $0.5 \pm 0.5$ years and estimate the maximum counting error (MCE) from the number of uncertain layers ($N$) as $N \times 0.5$ years. Using the records for four elements (i.e., Al, Ca, Fe, and Mg), we define "uncertain annual layer boundaries" as those without synchronous peaks of all four elements. Annual layer peaks for each of the three sections and their respective uncertainties are shown in Fig. 2. The number of annual layers for Section I, II, and III is $10 \pm 2$, $19 \pm 3$, and $23 \pm 3$. The derived average annual layer thickness for Section I, II, and III is thus $38.30 \pm^{9.57}_{6.38}$ mm (corresponding to $30.96 \pm^{7.74}_{5.16}$ mm w.e.), $18.42 \pm^{3.45}_{2.51}$ mm ($14.74 \pm^{2.76}_{2.01}$ mm w.e.), and $12.71 \pm^{1.91}_{1.47}$ mm ($10.16 \pm^{1.52}_{1.17}$ mm w.e.), respectively.

We also applied the StratiCounter program (Winstrup et al., 2012) to identify annual layers for Section III (Fig. S6). Despite minor differences, StratiCounter produced mostly comparable results. The average annual layer thickness derived from the StratiCounter counts is 12.60 mm (corresponding to 10.08 mm w.e.), which is roughly consistent with our estimate from the manual layer counting ($10.16 \pm^{1.52}_{1.17}$ mm w.e.). This confirms the reliability of our manual layer counting. The StratiCounter program could not be applied to Sections I and II due to their short duration. For consistency, we used results from manual layer counting for further analysis and discussion.

## 3.2  Thinning of annual layers due to ice flow

Glaciers consist of sequences of sedimentary deposits of annual snow in polar and alpine regions (Rapp, 2012). Snow layers sink into the ice mass and are subjected to continuous thinning (Nye, 1963; Rapp, 2012). This occurs initially due to densification, by which the snow is gradually transformed into ice, but later mainly because of flow-induced vertical compressive strain. Therefore, the observed thickness of an annual layer reflects both the initial amount of annual accumulation and the vertical compression the layer has been subject to since deposition (Paterson and Waddington, 1984). In this process, the ice layers are stretched horizontally until they are advected by the ice motion into an ablation zone (Rapp, 2012). The Chongce Core 2 was drilled on a flat platform with an area of over $160 \, \text{km}^2$ where the ice layers are horizontal. As a result, horizontal deformation has little effect on the redistribution of snow. In addition, temporal changes in basal topography are likely minimal due to the Holocene origin of the Chongce Core 2 (Hou et al., 2018; Licciulli et al., 2020). Therefore, here we estimated the vertical strain rate and accumulation rate of the Chongce Core 2 using a simple two-parameter steady-state flow model (Bolzan, 1985):

$$T_{(z)} = \frac{H}{bp} \left[ \left( 1 - \frac{z}{H} \right)^{-p} - 1 \right]. \tag{1}$$

The model has 2 degrees of freedom, the net annual accumulation rate $b$ and the thinning parameter $p$, both of which are assumed to be constant over time. $H$ is the glacier thickness (m w.e.). $z$ is the depth (m w.e.), and $T_{(z)}$ is the corresponding age at $z$. For the Chongce Core 2 (drilled to bedrock), $H$ is 112.243 m water equivalent (m w.e.), calculated as the product of the ice core length and its density. In order not to over emphasize the data by the deepest horizons, the model is fitted using the logarithms of the age values (Uglietti et al., 2016). The model was first constrained by the $^{14}$C calibrated ages, together with the $\beta$ activity reference horizon of the Chongce 58.82 m Core 3, located only $\sim 2$ m apart (Hou et al., 2018; Pang et al., 2020). We found that by using these data only, the model is poorly constrained at the deep section and gives an estimate bottom age much older than the bottom age ($8.3 \pm^{6.2}_{3.6}$ ka) estimated for Core 4 (Hou et al., 2018). Therefore, we included the Core 4 bottom age to constrain the final model. Due to its mathematical configuration to account for ice flow dynamics, the model gives more weight to points at shallower sections. Therefore, the inclusion of the Core 4 bottom age (relatively younger than otherwise derived bottom age) pushes the curve towards the left (younger) of most $^{14}$C dates (Fig. 3). The derived thinning parameter is 0.008 (dimensionless), and average annual accumulation of the entire ice core ($\sim 9$ ka to present) is $103 \pm 34$ mm w.e. (Fig. 3). This derived accumulation rate is in relative agreement with the average annual accumulation which was observed in the uppermost 50 annual layers ($\sim 140$ mm w.e. yr$^{-1}$), where the thinning effect is negligible (Hou et al., 2018). The derived ice age at the bedrock is $9.0 \pm^{7.9}_{3.6}$ ka, which is consistent with the bottom age ($8.3 \pm^{6.2}_{3.6}$ ka) estimated for Core 4. In addition, the modeled age at the depth of the oldest $^{14}$C sample is $5.2 \pm^{1.9}_{1.4}$ ka, similar to the actual $^{14}$C age of $6.3 \pm 0.2$ ka given the uncertainty range. Therefore, we believe this model gives the most reasonable results, compared with several other model fits based on different data combinations (Fig. S7). Details of these model fits are provided in Text S1 and Fig. S7.

## 3.3  Accumulation rates of the past four time windows

With the derived thinning parameter and average annual accumulation over the Holocene, we calculated the initial annual layer thickness (mm w.e.) for the average accumulation rate at various depths for the Top Section and Section I, II, and III using a simple flow model for the decrease of the annual layer thickness with depth (Bolzan, 1985; Uglietti et al., 2016):

$$L_{(z)} = b \left( 1 - \frac{z}{H} \right)^{1+p}, \tag{2}$$

where $L_{(z)}$ is the modeled annual layer thickness (mm w.e.) for the average accumulation rate ($b$, i.e., $103 \pm 34$ mm w.e.) at the depth of $z$ given the thinning parameter of $p$ (i.e., 0.008). The mean annual layer thickness derived from the

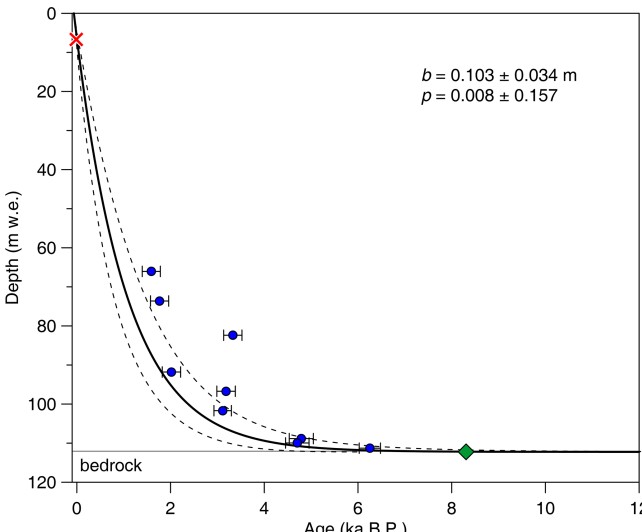

**Figure 3.** The depth–age relationship of the Chongce Core 2 based on the two-parameter model. The dashed lines represent the $1\sigma$ confidence interval of the two-parameter model fit (solid line). The red cross stands for the $\beta$ activity peak in 1963 CE, the blue dots the calibrated $^{14}$C ages with $1\sigma$ error bar, and the green diamond the bedrock age estimate from the Chongce Core 4 (Hou et al., 2018; Zhang et al., 2018).

ice flow model for the Top Section and Section I, II, and III are 99.11, 47.16, 20.75, and 9.61 mm w.e., respectively. To confirm the reliability of the dating results based on the ice flow model, we used Monte Carlo simulations (Breitenbach et al., 2012) to establish a continuous depth–age relationship of the Chongce Core 2 based on $^{14}$C ages and the $\beta$ activity horizon (An et al., 2016; Hou et al., 2018) (Fig. S8). This method was used before to establish the depth–age scale of the Mt. Ortles ice core extracted from the summit of Alto dell'Ortles in the Italian Alps (Gabrielli et al., 2016) and the Zangser Kangri ice core extracted from the northwestern Tibetan Plateau (Hou et al., 2021). This method can account for potential changes in snow accumulation and/or strain rate. The Monte Carlo-based annual layer thickness for the Top Section and Section I, II, and III are 122.04, 35.14, 40.79, and 9.06 mm w.e., respectively. For the Top Section, both the ice flow model and Monte Carlo simulations produced lower mean annual layer thickness than observed (140.76 mm w.e.). For Section I and II, both methods generated higher mean annual layer thicknesses than observed ($30.96\pm^{7.74}_{5.16}$ mm w.e. and $14.74\pm^{2.76}_{2.01}$ mm w.e.). For Section III, they produce comparable mean annual layer thicknesses to the observed value ($10.16\pm^{1.52}_{1.17}$ mm w.e.). Therefore, both methods produced relatively consistent temporal patterns, despite some difference in the results.

As this study aims to quantitatively reconstruct the annual accumulation from an alpine ice core, we focus on the annual layer thickness results derived from the ice flow model. The estimated original (pre-thinning) accumulation

(mm w.e.) for each annual layer can be derived through multiplying the ratio of the observed to modeled annual layer thickness (mm w.e.) by the average annual accumulation rate (103 mm w.e.) (Hou et al., 2018; Roberts et al., 2015; Winstrup et al., 2012). The observed layer thickness is established in Sect. 3.1. The results (Fig. 4) show that the mean annual accumulation was $108.86\pm^{16.29}_{12.54}$ mm w.e. at $\sim 2.5$ ka, which is comparable to the Holocene mean value. The mean annual accumulation was $73.73\pm^{13.80}_{10.05}$ mm w.e. at $\sim 1.7$ ka and $67.62\pm^{16.91}_{11.27}$ mm w.e. at $\sim 0.8$ ka, about 28 % and 34 % lower than the Holocene mean respectively. The mean annual accumulation was 146.30 mm w.e. during 1953–2012 CE, $\sim 42$ % higher than the Holocene mean.

## 4   Discussion

Alpine glaciers over the TP extend high into the middle troposphere, yielding ice cores that provide continuous annual accumulation records representative of a large area (Duan et al., 2015; Yao et al., 2008). However, not all snowfall is securely stored in high-elevation glaciers, due to wind scouring, snow drifting, and sublimation (Hardy et al., 2003). Moreover, a firnification process might develop rapidly, as indicated by the lack of lower-density layers (indicative of snow) near the glacier surface (Fig. S5). Therefore, ice core accumulation reconstructed in this paper is not a direct measurement of precipitation but rather a quantitative proxy of net precipitation in the western Kunlun Mountains. In Fig. 4, the reconstructed average annual accumulation of the four time windows was compared with other reconstructed and modeled precipitation series for the TP to evaluate the reliability of our method of reconstruction. Thompson et al. (1995) reconstructed a snow accumulation record for the last millennium from an ice core retrieved at the Guliya ice cap ($\sim 30$ km from the Chongce drilling site). Their reconstruction shows that snow accumulation rate for 1950–1989 CE (32.18 cm ice yr$^{-1}$) is 62 % higher than that for 1160–1169 CE (19.79 cm ice yr$^{-1}$) (Fig. 4b). This is largely consistent with the Chongce reconstruction, which shows a 65 % increase in the mean annual accumulation between 1160–1172 CE (84.28 mm w.e.) and 1953–1989 CE (139.03 mm w.e.). Yang et al. (2014) reconstructed annual precipitation over the past 3500 years using subfossil, archeological, and living juniper tree samples from the northeastern TP (Fig. 4c). Their reconstruction shows that the last 50 years is a very wet period relative to the past 3500 years, consistent with our reconstruction. In addition, both records show similar dry intervals in $\sim 0.8$ and $\sim 1.7$ ka and a moderately wet interval in $\sim 2.5$ ka (Fig. 4). Our reconstruction is also in agreement with the TraCE-21 ka model results (extracted for the study region of 34–36° N, 80–82° E), which simulate continuous climate evolution over the last 21 000 years (Collins et al., 2006) (Fig. 4d). These results suggest that the method proposed in this study has the poten-

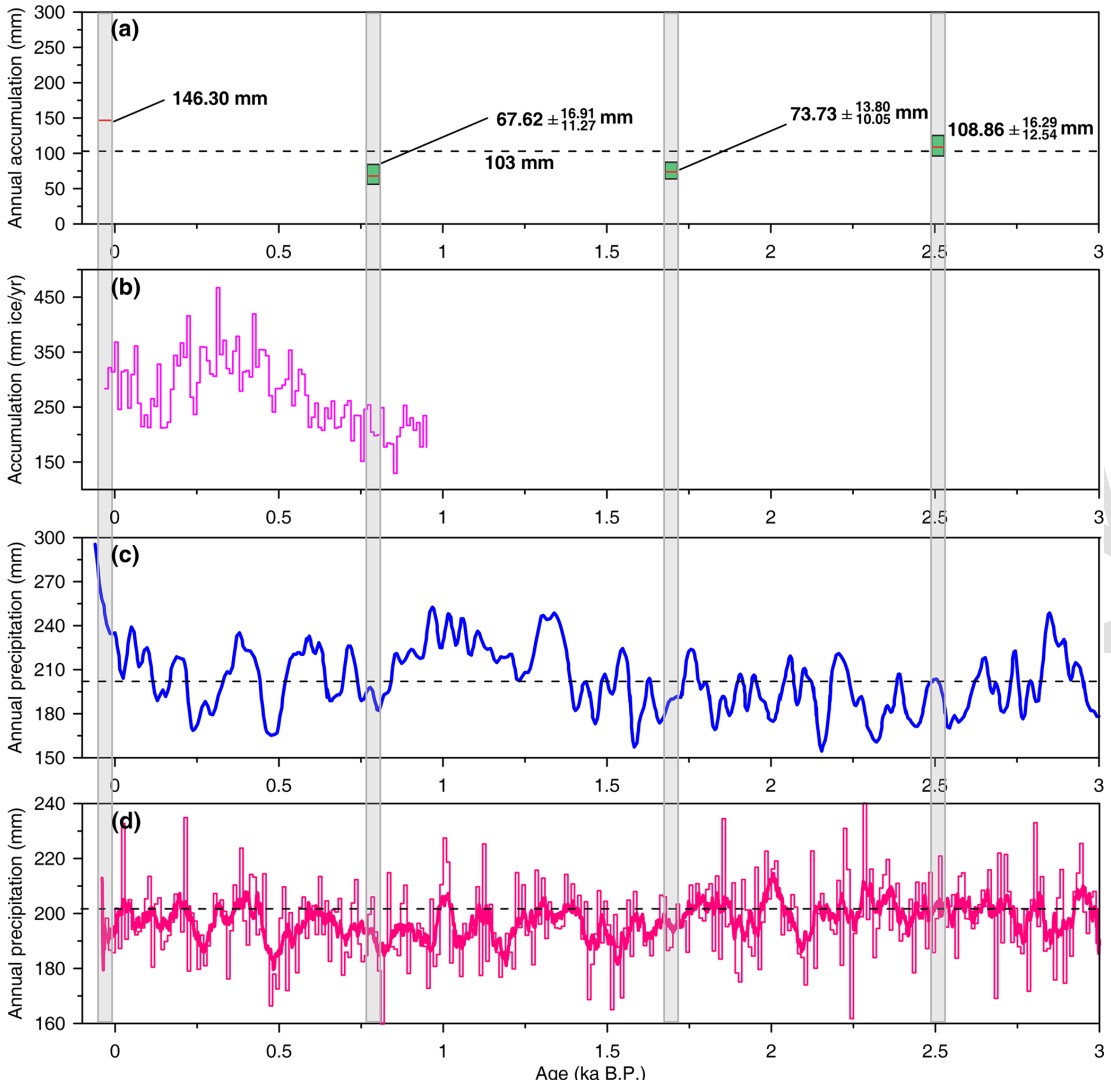

**Figure 4.** Our reconstructed annual accumulation for each of the four sections sampled **(a)** and its comparison with **(b)** snow accumulation reconstruction based on the Guliya ice core (Thompson et al., 2006), **(c)** precipitation reconstruction with 50-year smoothing in the northeastern TP (Yang et al., 2014), and **(d)** the TraCE-21 ka model results for the region (34–36° N, 80–82° E) (Collins et al., 2006). The dotted lines in **(a)** and **(d)** represent the average annual accumulation over the Holocene. The dotted line in **(c)** represents average annual accumulation over the past 3.0 ka. The thin and thick red lines in **(d)** represent the 10-year and 50-year moving average respectively.

tial to reconstruct high-resolution continuous precipitation time series.

Compared with the previous precipitation records based on paleoclimate proxies such as tree-ring width, pollen abundance index, ice core chemistry, and stalagmites, the method proposed in this paper has the significant advantage in quantifying annually resolved precipitation of past millennia. Previous quantitative precipitation reconstructions assume a stationary linear relationship between proxy data and actual precipitation over time (Tozer et al., 2016; Yang et al., 2014), which is difficult to establish over long periods. In comparison, net accumulation rates in ice cores provide more direct and quantitative data for past precipitation. The establish-

ment of a reliable annual accumulation record is determined only by two factors: (i) identification of the annual layers and (ii) the ice core thinning parameters.

The reliability of this method can be further improved with the existence of absolute chronological markers (e.g., volcanic events and archeological archives). If such markers exist in an ice core, a local average annual layer thickness ($b$) can be calculated between two adjacent markers by dividing the length of ice by the number of annual layers between them. The average net annual accumulation can then be determined from the ratio of the average net annual layer thickness (mm w.e.) to the flow-modeled thickness (mm w.e.) multiplied by the average annual accumulation of the Holocene.

This provides an additional way to verify the reconstructed accumulation rates. As a result, we can improve the reliability of our method by comparing three different reconstructions (i.e., the Holocene net mean accumulation derived from the two-parameter model, average net annual accumulation between adjacent absolute chronological markers, and average net annual accumulation of different time windows based on annual layer identification). In addition, we will perform more direct observations (e.g., surface and bedrock topography and borehole inclination angles) on the Chongce ice cap in the future and use them to constrain a state-of-the-art 3-D ice cap flow model (Licciulli et al., 2020). This could further improve our estimates of accumulation rate and the thinning parameter.

# 5 Conclusions

In this paper, we presented a quantitative method to reconstruct annual accumulation from alpine ice cores for past millennia, using as an example an ice core drilled at the Chongce ice cap in the northwestern TP. We used LA-ICP-MS technology to develop continuous ultra-high-resolution records of chemical constituents (Al, Ca, Fe, Na, Mg, Cu, and Pb) in three sections of the Chongce ice core (corresponding to 781–789, 1682–1697, and 2511–2541 years BP in age respectively). Based on the seasonality shown in these trace element records (Al, Ca, Fe, and Mg), we identified annual layers in each section. In addition, annual layers of the Top Section of Chongce ice core (1953–2012 CE in age) were identified based on seasonal cycles of $\delta^{18}O$ values. The thickness of these annual layers was subsequently corrected using a two-parameter flow model to establish initial net accumulation for these sections. The results show that the average annual accumulation was 109 mm around 2.5 ka, which is comparable to the Holocene average. The average accumulation was 74 mm at 1.7 ka and 68 mm at 0.8 ka, about 28 % and 34 % lower than the Holocene average. It reached a high value of 145 mm during 1953–2012 CE, about 41 % higher than the Holocene average. Our estimates are consistent with previous results from tree rings and the TraCE-21 ka transient model simulations. Therefore, the method has the potential to reconstruct continuous high-resolution precipitation records covering millennia or even longer time periods.

*Code and data availability.* The data used in this paper can be downloaded from the Zenodo repository at https://doi.org/10.5281/zenodo.4387022 (Zhang, 2020). Data of glaciers are from the Global Land Ice Measurements from Space (GLIMS; available at https://doi.org/10.7265/N5V98602, GLIMS and NSIDC, 2015). The code for the StratiCounter program is available at the GitHub repository (http://www.github.com/maiwinstrup/StratiCounter, last access: 14 September 2020; Winstrup, 2015; Winstrup et al., 2012).

*Supplement.* The supplement related to this article is available online at: https://doi.org/10.5194/tc-16-1-2022-supplement.

*Author contributions.* SH conceived this study and drilled the ice cores. WZ, TMJ, and EVK took the measurements. WZ wrote the paper. SYW, HP, SBS, PAM, TMJ, and MS helped analyze the results and revise the manuscript. All authors contributed to discussion of the results.

*Competing interests.* The contact author has declared that neither they nor their co-authors have any competing interests.

*Acknowledgements.* Many thanks are due to many scientists, technicians, graduate students, and porters, especially to Yongliang Zhang, Hao Xu, and Yaping Liu for their great efforts working at high elevations, to Mariusz Potocki for his help in measuring chemical constituents of ice core samples, to Chiara Uglietti and Heinz Walter Gäggeler for help in measuring the $^{14}C$ samples, and to Guocai Zhu for providing the ground-penetrating radar results. We are grateful to the two anonymous reviewers and the editor, Carlos Martin, for their constructive comments and suggestions, which helped to improve the paper.

*Financial support.* This research has been supported by the National Natural Science Foundation of China (grant nos. 42001050, 91837102, 41830644, and 42021001), the W. M. Keck Foundation, and the US National Science Foundation (grant nos. PLR-042883, PLR-1203640, and PLR-1417476).

*Review statement.* This paper was edited by Carlos Martin and reviewed by two anonymous referees.

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
