# Peer review of "A quantitative method of resolving annual precipitation for the past millennia from Tibetan ice cores"

_The Cryosphere, 2021_

## Referee Comment (RC1)

Comments to specific items of text are referened as Pxxx Lyyy for Page xxx Line yyy

**1 General Comments**

Zhang et al. "A new method of resolving annual precipitation for the past millennia from Tibetan ice cores" presents a detailed study on the average accumulation rate for 3 epochs in the last 2500 years for an ice core site on the Chongce ice cap, northwestern Tibetan Plateau. The paper combines annual layer thickness data (from ultra-high resolution ice core elemental chemistry) with a flow thinning model (constrained by water-insoluble organic carbon $^{14}$C ages) to determine local net accumulation over 3 disjoint epochs. The paper is well written and structured and generally presents sufficient supporting evidence. I recommend minor alterations and corrections detailed below.

**2 Specific Comments**

**2.1 Major Specific Comment**

The most significant problem with the manuscript as it stands is the data fit to the flow model presented in Figure 2 and associated text on P8 L228–229. In particular, it appears that all of the $^{14}$C ages are above the nonlinear least squares data fit. This raises questions about the validity of the data fit and if the solution has converged. I would have expected at least some of the $^{14}$C ages to be below the data fit. Specifically, the data fit line can be moved upward and this would reduce the error at every observational data point, and hence the overall error of the fit. The authors need to verify that the data fit presented is indeed a (near) optimal fit, and redo the accumulation analysis if the data fit needs to be revised and improved.

**2.2 Minor Specific Comments**

P2 L40 Is Christiansen and Ljungqvist (2017) the correct citation? This paper is about temperature reconstruction, and only mentions precipitation because of it's influence on temperature reconstructions.

P2 2nd paragraph. This needs a restructure, at the moment, the sentence topics are annual layers, thinning, annual layers then thinning again. Suggest you move the sentence staring "In addition, the nonlinear" to after the sentence staring "The most common approach". Then change "The thinning parameter" → "This thinning parameter".

P3 L80 I think the location map (Fig S1) should be moved into the main manuscript, as this is key information.

P5 Section 2.3 You do not give the vertical size of the samples required to give the 1kg sample, this is key information for the depth uncertainty estimate of the $\beta$-activity dating.

P5 L130 Was the Argon gas flow purged or was the system purged using Argon gas? If the later, suggest changing "whilst the Argon (Ar) gas flow was purged for two minutes" → "whilst the system was purged with Argon (Ar) gas for two minutes".

P5 Section 2.4 you do not give the vertical size of the samples used for the $^{14}$C extraction, this is key information for the uncertainty estimate of the $^{14}$C dating, as there is uncertainty in both the age and depth.

P6 L188–189 These grouped peaks could also be from independent snow events with dry wind blown dust deposition between these snow events.

P9 L241-242 Make it clear that you are using the values of "b" and "p" that you found in Section 3.2.

P10 L257 Change "can be securely stored" → "is preserved". In fact your density profiles (Fig S6) suggest this for Core 2 and 3, which both lack the lower densities near the surface indicative of snow. I suggest you add a sentence at Line 258 making this point.

P10 L265 You have presented all other accumulation rates as mm w.e./yr, suggest that you do the same for the Thompson et al (2006) results, to allow for easy comparison.

P10 L284–286 This statement is not correct. For example, an error in either the $^{14}$C dating, or the flow model fit (see main points above) will introduce an error in the flow thinning model, which due to its non-linear nature will result in different relative average accumulations over various epochs.

P11 L295–299 In fact you already have 9 such markers from the $^{14}$C age ties, which allow you to calculate the average accumulation rate over the 8 epochs these 9 makers define.

Supp info, Figure 1b Give details of where the remote sensing data is from, what is the instrument (e.g. optical, SAR) and give a data citation.

Supp info, Figure S8 Give details of which core (or cores) are being compared here.

Supp info, Table S1 is the depth in meters water equivalent? Explain the difference between "$^{14}$C age" and "cal age".

**3 Technical corrections**

P2 L34 Kidd and Hoffman 2011 do not say "most important" only "variable parameter associated with atmospheric circulation". Delete "most important".

P2 L45 "glacier" → "glaciers".

P2 L45–47 It is possible to obtain accumulation rates at time-scales other than annual from ice-cores. Suggest changing "obtain reliable annual-layer thickness information" → "obtain reliable layer thickness information for the relevant times-scales (typically annual, but may be centennial for low temporal resolution sites or studies)".

P2 L48 You are not constraining the thinning, you are compensating for it, suggest changing "constrained" → "compensated for".

P2 L57 There are many more ice core records than just the citations you list, suggest changing "(Alley" → "(e.g., Alley"

P3 L62 Change "methods. e.g., the" → "methods, for example the".

P3 L64 Remove the full stop after "technology".

P3 L65 Maybe change "reveal" to "resolve".

P3 L69 Change "parameters" → "parameterisations".

P3 L77 Delete the word "parameter".

P3 L78 Change "record" → "records".

P4 L93 See comment above about moving Fig S1 into the main manuscript.

P4 L96 Is there a citation for the statement that the local climate is "largely controlled by the mid-tropospheric westerlies"?

P4 L100 Given you have listed the summer (28%) and winter/spring (59%) precipitation percentages, also include for autumn (13%) rather than leave the reader to calculate this. Suggest changing "lowest amount of precipitation." → "lowest amount (13%) of precipitation."

P5 L145 There is some ambiguity about what you are removing the 3mm outer layer from, and while the reader can work it out, it is much better to make it easier for the reader to understand. Therefore, suggest changing "decontaminated the $^{14}$C samples" → "decontaminated the ice for the $^{14}$C samples".

P5 L147 The more common term is laminar flow "hood" rather than "box".

P6 L149 Delete "were".

P6 L153 Change "found in the previous studies (Uglietti et al., 2016)." → "found in Uglietti et al. (2016)."

P6 S2.5 You talk about verifying your annual-layer identification using Strati-Counter, but at this point in the manuscript you haven't described how you did your annual-layer identification. As this description comes later, suggest changing "To verify our annual-layer identifications" → "To verify our annual-layer identifications (see Section 3.1)".

P6 L166 While CCSM3 might have been "state-of-the-art" when this research was conducted (2006), this is no longer the case, with CCSM3 being replaced by CCSM4 in 2010. Suggest deleting "state-of-the-art".

P8 L208-209 Until this point your references have been in alphabetic order, so suggest you swap order of Rapp 2012 and Nye 1963.

P8 L225 Change "overweigh" → "over emphasise".

P9 L236 Change "of the Holocene" → "over the Holocene".

P9 L245 Change "The initial" → "The estimated original (pre-thinning)".

P13 L344 Change "Bronk Ramsey, C.," → "Ramsey, C.B.,".

P13 L350 Delete second, repeated "for large-scale temperature".

P15 L412 I don't think Parrenin et al 2004 is cited anywhere in the manuscript.

P16 L443 I don't think Tang et al 2015 is cited anywhere in the manuscript.

P17 L475 Change "sine" → "since"

Supp info, Figure S4 Change "The seasonal precipitation" → "Monthly precipitation".

---

## Author Comment (AC1)

Responses to Comments on the Manuscript:

"**A new method of resolving annual precipitation for the past millennia from Tibetan ice cores**"

(MS No.: tc-2021-115)

Many thanks for the reviewer's constructive comments. Below a point-to-point response to the comments. The comments are in black, and our response is in blue.

**1 General Comments**

Zhang et al. "A new method of resolving annual precipitation for the past millennia from Tibetan ice cores" presents a detailed study on the average accumulation rate for 3 epochs in the last 2500 years for an ice core site on the Chongce ice cap, northwestern Tibetan Plateau. The paper combines annual layer thickness data (from ultra-high resolution ice core elemental chemistry) with a flow thinning model (constrained by water-insoluble organic carbon [14]C ages) to determine local net accumulation over 3 disjoint epochs. The paper is well written and structured and generally presents sufficient supporting evidence. I recommend minor alterations and corrections detailed below.

**2 Specific Comments**

**2.1 Major specific comment**

The most significant problem with the manuscript as it stands is the data fit to the flow model presented in Figure 2 and associated text on P8 L228-229. In particular, it appears that all of the [14]C ages are above the nonlinear least squares data fit. This raises questions about the validity of the data fit and if the solution has converged. I would have expected at least some of the [14]C ages to be below the data fit. Specially, the data fit line can be moved upward and this would reduce the error at every observational data point, and hence the overall error of the fit. The authors need to verify that the data fit presented is indeed a (near) optimal fit, and redo the accumulation analysis if the data fit needs to be revised and improved.

Response: In this study, the depth-age relationship of the Chongce 135.81 m Core 2 was established by using a two-parameter (2p) model. The 2p model was first constrained by the [14]C calibrated ages, together with the $\beta$-activity reference horizon of the Chongce 58.82 m Core 3, located only ~ 2 meters apart (Hou et al., 2018; Pang et al., 2020). We found that by using these data only, the 2p model is poorly constrained at the deep section, and giving an estimate bottom age much older than the bottom age ($8.3 \pm_{3.6}^{6.2}$ ka B.P.) estimated for Core 4 (Hou et al., 2018). Therefore, we included the Core 4 bottom age to constrain the final 2p model. Due to its mathematical configuration to account for ice flow dynamics, the 2p model gives more weight to points at deeper sections. Therefore, the inclusion of the Core 4 bottom age (relatively younger than otherwise derived bottom age) pushes the curve towards the left (younger) of most [14]C dates. However, we believe this model gives the most reasonable results, compared with several other model fit based on different data combinations (Figure 1). The details of these model fits are provided as follows.

(1) all data (including $\beta$-activity peak of Core 3 and nine [14]C ages) (Fig. 1a).

Results: The derived annual accumulation rate of $137 \pm 54$ mm w.e./year is in good agreement with the value of 140 mm w.e./year based on the tritium horizon. But the model is poorly constrained in deeper sections: the derived age estimate at the depth of the deepest [14]C sample is $9.1 \pm_{4.0}^{7.2}$ ka B.P.. This is much older than the actual measured [14]C age of $6.3 \pm 0.2$ ka B.P. at that depth (Fig. 1a).

(2) all data (including $\beta$-activity peak of Core 3 and nine [14]C ages) and constant accumulation rate (140 m w.e./year) (Fig. 1b).

Results: The derived ice age at the bedrock is $30.7 \pm_{18.4}^{44.8}$ ka B.P., which is much older than the bottom age ($8.3 \pm_{3.6}^{6.2}$ ka B.P.) estimated for Core 4. In addition, the derived age estimate at the depth of the deepest [14]C sample

is $9.2 \pm_{3.6}^{6.0}$ ka B.P.. This is much older than the $^{14}$C age of $6.3 \pm 0.2$ ka B.P. at that depth. (Fig. 1b).

(3) $\beta$-activity peak of Core 3 and oldest six $^{14}$C ages (Fig. 1c).

Results: The derived ice age at the bedrock is $22.5 \pm_{13.8}^{34.8}$ ka B.P., which is much older than the bottom age $(8.3 \pm_{3.6}^{6.2}$ ka B.P.) estimated for Core 4. In addition, the derived accumulation ($233 \pm 104$ mm w.e./year) deviates significantly from the $\beta$-activity based estimate (140 mm w.e./year) (Fig. 1c).

(4) $\beta$-activity peak of Core 3, oldest six $^{14}$C ages, and constant accumulation rate (140 mm w.e./year) (Fig. 1d).

Results: The derived ice age at the bedrock is $50.1 \pm_{35.6}^{118.4}$ ka B.P., which is much older than the bottom age $(8.3 \pm_{3.6}^{6.2}$ ka B.P.) estimated for Core 4. In addition, the derived age estimate at the depth of the deepest $^{14}$C sample is $9.6 \pm_{4.1}^{7.3}$ ka B.P.. This is much older than the $^{14}$C age of $6.3 \pm 0.2$ ka B.P. at that depth (Fig. 1d).

(5) all data (including $\beta$-activity peak of Core 3 and nine $^{14}$C ages) plus bedrock estimate from Core 4 (Hou et al., 2018) as an additional model input point (**the method used in this manuscript**) (Fig. 1e).

Results: The derived ice age at the bedrock is $9.0 \pm_{3.6}^{7.9}$ ka B.P., which is roughly consistent with the bottom age $(8.3 \pm_{3.6}^{6.2}$ ka B.P.) estimated for Core 4. The derived accumulation rate ($103 \pm 34$ mm w.e./year) is in relative agreement with the $\beta$-activity based estimate (140 mm w.e./year). In addition, the modeled age at the depth of the deepest $^{14}$C sample is now $5.2 \pm_{1.2}^{1.9}$ ka B.P. which, with the uncertainty range, is similar to the $^{14}$C age of $6.3 \pm 0.2$ ka B.P. (Fig. 1e). We believe this model provides most reasonable results, and is therefore adopted for this paper.

[Figure]

Fig.1. The depth-age relationship of the Chongce Core 2 based on the two-parameter model.

**2.2 Minor specific comment**

P2 L40 Is Christiansen and Ljungqvist (2017) the correct citation? This paper is about temperature reconstruction, and only mentions precipitation because of its influence on temperature reconstructions.

Response: We replaced this citation with Sun et al., 2018, which presented a comprehensive review of the data sources

and estimation methods of 30 currently available global precipitation data sets, including gauge-based, satellite-related, and reanalysis data sets.

P2 2nd paragraph. This needs a restructure, at the moment, the sentence topics are annual layers, thinning, annual layers then thinning again. Suggest you move the sentence starting "In addition, the nonlinear" to after the sentence starting "The most common approach". Then change "The thinning parameter" → "This thinning parameter".

Response: We agree with the reviewer, and have revised the sentence accordingly. The revised sentence is as follows; *The most common approach is to obtain annual-layer thickness based on the seasonal cycles of ice core parameters such as stable isotope ratio of oxygen in the water ($\delta^{18}O$), the concentration of major ions (e.g. $Ca^{2+}$, $Mg^{2+}$, $NH_4^+$, $SO_4^{2-}$), and the presence of melt layers (Thompson et al., 2018). In addition, the nonlinear thinning of annual layers caused by ice flow must be suitably constrained (Bolzan, 1985; Henderson et al., 2006; Roberts et al., 2015).*

P3 L80 I think the location map (Fig. S1) should be moved into the main manuscript, as this is key information.
Response: We agree with the reviewer, and have included the location map (Fig. S1) in the main text of the manuscript.

P5 Section 2.3 You do not give the vertical size of the samples required to give the 1kg sample, this is key information for the depth uncertainty estimate of the $\beta$-activity dating.
Response: We have given details on ice samples for $\beta$-activity measurements (Table S1) in the supporting information.

Table S1. Details on ice samples for $\beta$-activity measurements.

| Sample # | Depth (m) | Depth (m w.e.) | Length (m) | Length (m w.e.) | $\beta$ activity (dph kg$^{-1}$) |
|---|---|---|---|---|---|
| 1 | 0.000-0.710 | 0.000-0.406 | 0.710 | 0.406 | 555.1 |
| 2 | 0.710-1.150 | 0.406-0.771 | 0.440 | 0.365 | 936.5 |
| 3 | 1.150-1.720 | 0.771-1.253 | 0.570 | 0.482 | 597.9 |
| 4 | 1.720-2.185 | 1.253-1.648 | 0.465 | 0.395 | 499.2 |
| 5 | 2.185-2.575 | 1.648-1.981 | 0.390 | 0.333 | 505.6 |
| 6 | 2.575-2.945 | 1.981-2.297 | 0.370 | 0.316 | 539.1 |
| 7 | 2.945-3.355 | 2.297-2.648 | 0.410 | 0.351 | 416.7 |
| 8 | 3.355-3.890 | 2.648-3.110 | 0.535 | 0.462 | 518.4 |
| 9 | 3.890-4.350 | 3.110-3.504 | 0.460 | 0.393 | 396.1 |
| 10 | 4.350-4.805 | 3.504-3.889 | 0.455 | 0.385 | 439.4 |
| 11 | 4.805-5.270 | 3.889-4.288 | 0.465 | 0.399 | 1754.5 |
| 12 | 5.270-5.780 | 4.288-4.735 | 0.510 | 0.447 | 385.8 |
| 13 | 5.780-6.320 | 4.735-5.198 | 0.540 | 0.463 | 504.9 |
| 14 | 6.320-6.780 | 5.198-5.593 | 0.460 | 0.395 | 749.1 |
| 15 | 6.780-7.200 | 5.593-5.948 | 0.420 | 0.355 | 963.2 |
| 16 | 7.200-7.690 | 5.948-6.362 | 0.490 | 0.414 | 224.9 |
| 17 | 7.690-8.170 | 6.362-6.767 | 0.480 | 0.406 | 1709.9 |
| 18 | 8.170-8.630 | 6.767-7.158 | 0.460 | 0.390 | 1910.3 |
| 19 | 8.630-9.120 | 7.158-7.571 | 0.490 | 0.413 | 479.9 |
| 20 | 9.120-9.580 | 7.571-7.977 | 0.460 | 0.407 | 574.2 |
| 21 | 9.580-10.020 | 7.977-8.361 | 0.440 | 0.384 | 98.6 |
| 22 | 10.020-10.550 | 8.361-8.819 | 0.530 | 0.457 | 682.8 |
| 23 | 10.550-11.060 | 8.819-9.254 | 0.510 | 0.435 | 262.6 |

| | | | | | |
|---|---|---|---|---|---|
| 24 | 11.060-11.490 | 9.254-9.618 | 0.430 | 0.364 | 503.8 |
| 25 | 11.490-12.015 | 9.618-10.061 | 0.525 | 0.444 | 705.8 |
| 26 | 12.015-12.525 | 10.061-10.494 | 0.510 | 0.433 | 168.7 |
| 27 | 12.525-12.925 | 10.494-10.833 | 0.400 | 0.339 | 282.9 |
| 28 | 12.925-13.375 | 10.833-11.203 | 0.450 | 0.370 | 191.8 |
| 29 | 13.375-13.845 | 11.203-11.608 | 0.470 | 0.405 | 673.8 |
| 30 | 13.845-14.305 | 11.608-11.999 | 0.460 | 0.392 | 269.3 |
| 31 | 14.305-14.805 | 11.999-12.410 | 0.500 | 0.411 | 324.3 |

P5 L130 Was the Argon gas flow purged or was the system purged using Argon gas? If the later, suggest changing "whilst the Argon (Ar) gas flow was purged for two minutes" → "whilst the system was purged with Argon (Ar) gas for two minutes".

Response: We thank the reviewer for clarification, and have revised this sentence accordingly, as "whilst the system was purged with Argon (Ar) gas for two minutes".

P5 Section 2.4 you do not give the vertical size of the samples used for the [14]C extraction, this is key information for the uncertainty estimate of the [14]C dating, as there is uncertainty in both the age and depth.

Response: We have given the vertical size of the samples used for the [14]C extraction in the supporting information.

P6 L188-189 These grouped peaks could also be from independent snow events with dry wind blown dust deposition between these snow events.

Response: We agree with the reviewer, and have revised the text accordingly. The revised sentence is as follows;
*These grouped peaks are interpreted as independent snow events with elevated element concentrations or with wind-blown dust deposition between these snow events.*

P9 L241-242 Make it clear that you are using the values of "*b*" and "*p*" that you found in Section 3.2.

Response: We have revised this sentence as "where $L_{(Z)}$ is the modeled annual layer thickness (mm w.e.) for the average accumulation rate (*b*, i.e., $103 \pm 34$ mm w.e.) at the depth of *z* given the thinning parameter of *p* (i.e., 0.008).".

P10 L257 Change "can be securely stored" → "is preserved". In fact your density profiles (Fig. S6) suggest this for Core 2 and 3, which both lack the lower densities near the surface indicative of snow. I suggest you add a sentence at Line 258 making this point.

Response: Following the reviewer's comment, we have revised this sentence as "However, not all snowfall is preserved in high-elevation glaciers, due to wind scouring, snow drifting, and sublimation (Hardy et al., 2003). Moreover, firnification process might develop rapidly as indicated from the lack the lower density layers (indicative of snow) near the glacier surface (Fig. S6)".

P10 L265 You have presented all other accumulation rates as mm w.e./yr, suggest that you do the same for the Thompson et al (2006) results, to allow for easy comparison.

Response: We agree with the reviewer, but because the density profile of the Guliya ice core is not available (Thompson et al., 1995), we are not able to calculate the accumulation rate of the Guliya ice core as mm w.e./yr, but this comparison is still reasonable given the similar density for the periods of 1950-1989 A.D. and 1160-1169 A.D.

P10 L284-286 This statement is not correct. For example, an error in either the [14]C dating, or the flow model fit (see main

points above) will introduce an error in the flow thinning model, which due to its non-linear nature will result in different relative average accumulations over various epochs.

Response: We agree with the reviewer. For this reason, we deleted this statement in the revision.

P11 L295-299 In fact you already have 9 such markers from the $^{14}$C age ties, which allow you to calculate the average accumulation rate over the 8 epochs these 9 makers define.

Response: This suggestion is theoretically possible, but we are not able to calculate the average accumulation rates over the 8 epochs between the 9 $^{14}$C age ties because the errors of the $^{14}$C ages cause overlaps of some ages.

Supp info, Figure 1b give details of where the remote sensing data is from, what is the instrument (e.g. optical, SAR) and give a data citation.

Response: We have included details about the remote sensing data, and provided a citation in the supporting information.

Supp info, Figure S8 Give details of which core (or cores) are being compared here.

Response: We have included details of the ice cores in the supporting information.

Supp info, Table S1 is the depth in meters water equivalent? Explain the difference between "$^{14}$C age" and "cal age".

Response: The depth in Table S2 is the measured depth in the field. For convenience of the readers, we also included the depth in meters water equivalent in the revision after taking account of the density profile.

Regarding "$^{14}$C age" and "cal age", "$^{14}$C age" denotes conventional radiocarbon age, which is calculated from the formula below:

$$t = -8033 \times \ln (Fs)$$

where $t$ is conventional radiocarbon age, Fs is the $^{14}$C / $^{12}$C ratio of the sample divided by the same ratio of the modern standard. "cal age" denotes the calibrated age using OxCal v4.3 (Ramsey and Lee, 2013) with the Northern (IntCal13) calibration curve.

Table S2. Results of radiocabon measurements for the Chongce 135.81 m Core 2 ice core samples. For the calibrated calender year, ranges are given with 68.2% probality.

| Sample # | Depth (m) | Depth (m w.e.) | Mass (g) | WIOC (μg) | F$^{14}$C | $^{14}$C age (ka B.P.) | Calibrated age (ka B.P.) |
|---|---|---|---|---|---|---|---|
| CC-1 | 79.46-80.21 | 65.74-66.31 | 307.7 | 20.3 ± 1.2 | 0.81 ± 0.01 | 1.679 ± 0.078 | 1.445-1.704 |
| CC-2 | 88.82-89.56 | 73.31-73.92 | 302.9 | 24.3 ± 1.4 | 0.80 ± 0.01 | 1.831 ± 0.138 | 1.572-1.921 |
| CC-3 | 99.44-100.10 | 82.12-82.65 | 304.6 | 13.8 ± 0.9 | 0.68 ± 0.01 | 3.133 ± 0.161 | 3.157-3.560 |
| CC-4 | 110.58-111.35 | 91.48-92.10 | 342.6 | 24.9 ± 1.4 | 0.78 ± 0.01 | 2.037 ± 0.142 | 1.827-2.296 |
| CC-5 | 116.62-117.43 | 96.39-97.05 | 330.9 | 9.1 ± 0.7 | 0.69 ± 0.01 | 3.012 ± 0.164 | 2.978-3.377 |
| CC-6 | 122.64-123.36 | 101.40-101.98 | 338.6 | 17.6 ± 1.1 | 0.69 ± 0.01 | 2.944 ± 0.157 | 2.892-3.331 |
| CC-7 | 131.41-132.10 | 108.54-109.12 | 324.6 | 22.6 ± 1.3 | 0.59 ± 0.01 | 4.228 ± 0.176 | 4.451-5.036 |
| CC-8 | 132.65-133.51 | 109.59-110.31 | 392.7 | 23.6 ± 1.4 | 0.60 ± 0.01 | 4.169 ± 0.175 | 4.424-4.951 |
| CC-9 | 134.31-135.03 | 110.98-111.59 | 292.4 | 23.0 ± 1.4 | 0.51 ± 0.01 | 5.466 ± 0.201 | 5.997-6.443 |

**3 Technical corrections**

P2 L34 Kidd and Hoffman 2011 do not say "most important" only "variable parameter associated with atmospheric circulation". Delete "most important".

Response: Correction has been made accordingly.

P2 L45 "glacier" → "glaciers".

Response: Change has been made accordingly.

P2 L45-47 It is possible to obtain accumulation rates at time-scales other than annual from ice-cores. Suggest changing "obtain reliable annual-layer thickness information"→"obtain reliable layer thickness information for the relevant times-scales (typically annual, but may be centennial for low temporal resolution sites or studies)".

Response: Change has been made accordingly.

P2 L48 You are not constraining the thinning, you are compensating for it, suggest changing "constrained" → "compensated for".

Response: Change has been made accordingly.

P2 L57 There are many more ice core records than just the citations you list, suggest changing "(Alley" → "(e.g., Alley"

Response: Change has been made accordingly.

P3 L62 Change "methods. e.g., the" → "methods, for example the".

Response: Change has been made accordingly.

P3 L64 Remove the full stop after "technology".

Response: Change has been made accordingly.

P3 L65 Maybe change "reveal" to "resolve".

Response: Change has been made accordingly.

P3 L69 Change "parameters" → "parameterisations"

Response: Change has been made accordingly.

P3 L77 Delete the word "parameter".

Response: Change has been made accordingly.

P3 L78 Change "record" → "records"

Response: Change has been made accordingly.

P4 L93 See comment above about moving Fig S1into the main manuscript.

Response: Change has been made accordingly.

P4 L96 Is there a citation for the statement that the local climate is "largely controlled by the mid-tropospheric westerlies"?

Response: Yes, we have added a citation (i.e., Pang et al. (2020)).

P4 L100 Given you have listed the summer (28%) and winter/spring (59%) precipitation percentages, also include for autumn (13%) rather than leave the reader to calculate this. Suggest changing "lowest amount of precipitation." → "lowest amount (13%) of precipitation."

Response: Change has been made accordingly.

P5 L145 There is some ambiguity about what you are removing the 3mm outer layer from, and while the reader can work it out, it is much better to make it easier for the reader to understand. Therefore, suggest changing "decontaminated the $^{14}$C samples" → "decontaminated the ice for the $^{14}$C samples".
Response: Change has been made accordingly.

P5 L147 The more common term is laminar flow "hood" rather than "box".
Response: Change has been made accordingly.

P6 L149 Delete "were"
Response: Change has been made accordingly.

P6 L153 Change "found in the previous studies (Uglietti et al., 2016)." → "found in Uglietti et al. (2016). ".
Response: Change has been made accordingly.

P6 S2.5 You talk about verifying your annual-layer identification using StratiCounter, but at this point in the manuscript you haven't described how you did your annual-layer identification. As this description comes later, suggest changing "To verify our annual-layer identifications" → "To verify our annual-layer identifications (see Section 3.1)".
Response: Change has been made accordingly.

P6 L166 While CCSM3 might have been "state-of-the-art" when this research was conducted (2006), this is no longer the case, with CCSM3 being replaced by CCSM4 in 2010. Suggest deleting "state-of-the-art".
Response: Change has been made accordingly.

P8 L208-209 Until this point your references have been in alphabetic order, so suggest you swap order of Rapp 2012 and Nye 1963.
Response: Change has been made accordingly.

P8 L225 Change "overweigh" → "over emphasise".
Response: Change has been made accordingly.

P9 L236 Change "of the Holocene" → "over the Holocene".
Response: Change has been made accordingly.

P9 L245 Change "The initial" → "The estimated original (pre-thinning)".
Response: Change has been made accordingly.

P13 L344 Change "Bronk Ramsey, C.," → "Ramsey, C. B.,".
Response: Change has been made accordingly.

P13 L350 Delete second, repeated "for large-scale temperature".
Response: Change has been made accordingly.

P15 L412 I don't think Parrenin et al 2004 is cited anywhere in the manuscript.

Response: We have deleted this citation in the revision.

P16 L443 I don't think Tang et al 2015 is cited anywhere in the manuscript.

Response: We have deleted this citation in the revision.

P17 L475 Change "sine" → "since".

Response: Change has been made accordingly.

Supp info, Figure S4 Change "The seasonal precipitation" → "Monthly precipitation".

Response: Change has been made accordingly.

**References**

Bolzan, J. F.: Ice flow at the Dome C ice divide based on a deep temperature profile, J. Geophys. Res., 90(D5), 8111–8124, https://doi.org/10.1029/JD090iD05p08111, 1985.

Hardy, D. R., Vuille, M., and Bradley, R. S.: Variability of snow accumulation and isotopic composition on Nevado Sajama, Bolivia. J. Geophys. Res, 108(D22), https://doi.org/10.1029/2003JD003623, 2003.

Henderson, K., Laube, A., Gäggeler, H. W., Olivier, S., Papina, T., and Schwikowski, M.: Temporal variations of accumulation and temperature during the past two centuries from Belukha ice core, Siberian Altai, J. Geophys. Res., 111, D03104, https://doi.org/10.1029/2005JD005819, 2006.

Hou, S., Jenk, T. M., Zhang, W., Wang, C., Wu, S., Wang, Y., Pang, H., and Schwikowski, M.: Age ranges of the Tibetan ice cores with emphasis on the Chongce ice cores, western Kunlun Mountains, The Cryosphere, 12, 2341–2348, https://doi.org/10.5194/tc-12-2341-2018, 2018.

Ramsey, C. B., and Lee, S.: Recent and planned developments of the program Oxcal, Radiocarbon, 55, 720–730, 2013.

Roberts, J., Plummer, C., Vance, T., van Ommen, T., Moy, A., Poynter, S., Treverrow, A., Curran, M., and George, S.: A 2000-year annual record of snow accumulation rates for Law Dome, East Antarctica, Clim. Past, 11, 697–707, https://doi.org/10.5194/cp-11-697-2015, 2015.

Pang, H., Hou, S., Zhang, W., Wu, S., Jenk, T. M., Schwikowski, M., and Jouzel, J.: Temperature Trends in the Northwestern Tibetan Plateau Constrained by Ice Core Water Isotopes Over the Past 7,000 Years, J. Geophys. Res. Atmos., 125(19), e2020JD032560, 2020.

Sun, Q., Miao, C., Duan, Q., Ashouri, H., Sorooshian, S., and Hsu, K.-L.: A review of global precipitation data sets: Data sources, estimation, and intercomparisons, Rev. Geophys., 56, 79–107, https://doi.org/10.1002/2017RG000574, 2018.

Thompson, L., Mosley-Thompson, E., Brecher, H., Davis, M., León, B., Les, D., Lin, P.-N., Mashiotta, T., and Mountain, K.:Abrupt tropical climate change: Past and present, P. Natl. Acad. Sci. USA, 103(28), 10536-10543, https://doi.org/10.1073/pnas.0603900103, 2006.

Thompson, L. G., Mosley-Thompson, E., Davis, M. E., Lin, P. N., Dai, J., and Bolzan, J. F.: A 1000 year climate ice-core record from the Guliya ice cap, China: its relationship to global climate variability, Ann. Glaciol., 21, 175–181, https://doi.org/ 10.1017/S0260305500015780, 1995.

---

## Author Comment (AC2)

Responses to Comments on the Manuscript:
**"A new method of resolving annual precipitation for the past millennia from Tibetan ice cores"**
(MS No.: tc-2021-115)

We sincerely thank Prof. Jihong Cole-Dai for his thoughtful comments. Below we have made point-to-point responses to the comments. The comments are in black, and our responses in blue.

The paper's main objective is to reconstruct precipitation records from ice cores. The steps to accomplish the objective (Linea 80-88) are (1) determining thickness of annual layers in ice cores, (2) modeling layer thinning caused by ice flow, and (3) combining the results of (1) and (2) to reconstruct past precipitation. The authors call this process a "new method" to develop annual precipitation records from ice cores.

I am not sure what is new in the items presented in the paper. For Step 1, the way to identify annual layers in this work is the use of measurement of chemical impurities in ice cores that show annual oscillations in amount or concentration. But this is not new. Numerous papers have documented the identification of annual layers in ice cores using various chemical species measured at reasonable or acceptable temporal resolution and the use of the identified annual layers to derive layer thickness and the number of annual layers in a core (dating). Perhaps the authors mean that the chemical analysis technique (LA-ICP-MS) is new to ice cores. But a measurement technique (elemental measurement) for Ca different from the other techniques (e.g., ion chromatography) does not necessarily make the method to identify annual layers new. For example, when hydrogen peroxide was first measured in ice cores and its concentration was found to oscillate annually, $H_2O_2$ measurement for annual layer counting was not considered a new method of dating or annual layer determination (Sigg and Neftel, 1980 and Sigg et al., 1994). Maybe the authors believe the millimeter LA-ICP-MS analysis resolution is new; but measurements such as electric conductivity (ECM) and dielectric profiling (DEP), which have been used for quite a while, can also be of high-resolution (mm or sub-mm).

Response: Although ice core annual layer identification and layer thinning modeling are not new, they have not been used to reconstruct detailed (e.g. annual) precipitation time series beyond a few tens of years from Tibetan ice cores due to technical challenges and limitations. The new method presented in study has the potential to develop annual precipitation time series at millennial time scale from alpine ice cores, which has not been achieved so far from any previous studies.

We agree with Prof. Cole-Dai that numerous papers have documented the identification of annual layers in ice cores using various chemical species measured at reasonable or acceptable temporal resolution. However, due to rapid layer thinning, annual layer identification was only possible for the top sections (meters or tens meters) of the Tibetan ice cores from the conventional discrete measurements (e.g., water stable isotopes, major ions, dust) used in these studies, with a typical sampling resolution of centimeters. Although ECM and DEP can be of high-resolution (mm or sub-mm) and have been widely used for polar ice cores (Rasmussen et al., 2006), both techniques have not been used for the Tibetan ice cores to establish their chronology. This is largely because the high dust background of these ice cores causes a high noise/signal ratio that obscures the seasonality of their ECM and DEP profiles.

Our study is indeed the first attempt to use LA-ICP-MS on a Tibetan ice core. Given the ultra-high sampling resolution of LA-ICP-MS (153 $\mu$m per sample in our study), two orders of magnitude higher than the sampling resolution (centimeters) of the conventional chemical measurements, this technique allows us to identify annual cycles in deeper sections of the Chongce ice core (back to 2.5 ka B.P. in this study, in comparison to the usual tens years based on the conventional discrete measurements).

It is also worth pointing out that the [14]C measurements were first applied to establish the chronology of Tibetan Chongce ice cores (Hou et al., 2018, 2019 and 2021). This provides the necessary information of layer thinning to

develop the new method. We wish that this method will be used to reconstruct a continuous high-resolution precipitation record at millennial time scale when LA-ICP-MS is performed on the entire Tibetan ice cores.

The paper title suggests that the authors think this is the first time ("new") that annual layer identification using high-resolution chemical measurement is applied to mountain glacier ice cores. But that point is not apparent or explicit in the paper body.

Response: It is indeed the first time that the ultra-high-resolution LA-ICP-MS measurement is applied to mountain glacier ice cores, making it possible to identify annual layers at deeper sections of these cores. It is also the first time that such annual layer identification is combined with layer thinning modeling to reconstruct annual precipitation records at the millennial time scale from the Tibetan ice cores. This new method has potential to be applied to other alpine ice cores. We have clarified these points in the revision.

For Step 2, the modeling of ice flow including the method to quantify layer thinning rate is obviously not new. So, reconstruction of original layer thickness using measured layer thickness and modeled thinning curve should not be considered new, if the layer identification method is not new. In fact, reconstruction of accumulation records from measured annual layer thickness and modeled and experimentally determined layer thinning has been used frequently for polar ice cores, with records as long as tens of thousands of years. Yet, the authors seem to suggest (Lines 59-60: "challenging to develop annually resolved accumulation records covering longer (e.g. millennial) time periods") that this is a rare accomplishment. Again, this would sound reasonable when referring to mountain glacier ice cores (but not so for polar ice cores).

Response: We agree with the reviewer that the novelty of the method only applies to mountain glacier ice cores. Annual layer identification can be achieved for the Greenland summit ice cores for the past tens of thousands of years (Rasmussen et al., 2006), but this is not the case regarding the mountain glacier ice cores due to their relatively short length and rapid thinning. Our study is indeed the first to apply this method, i.e. taking account of both annual layer identification at the millennial time scale and annual layer thinning, for the Tibetan ice cores. It has also potential to be applied to other alpine ice cores. We have clarified these points in the revision.

The authors state (Lines 187-198) that identification of annual layers using concentration cycles of the selected elements was "verified" with counting by the StratiCounter program. But the StratiCounter algorithm is not designed to count layers accurately. It is meant to facilitate the counting process and make counting less subjective. Therefore, counting with StratiCount ought not to be used to verify manual counting. In fact, the algorithm is supposed to be used only when the annual signals are clear and consistent (i.e., highly unambiguous; Sigl et al., 2015). In this case, with many annual signals highly ambiguous (multiple peaks in one annual layer; see Figure 1), result of Straticounting cannot be viewed as verification of the result of manual counting.

Response: We agree with the reviewer's concern on the StratiCounter program. Indeed, in this study, we relied on visual identification and manual counting as our primary method, using StratiCounter only for comparison purposes. In our case, the results from manual counting and StratiCounter are highly consistent, reinforcing the validity of our results. We revised the wording in the manuscript to clarify this.

The precision (i.e., uncertainty) of annual layer counting (ALC) is critically important to the success of layer thickness determination and ice core dating. Often times, researchers compare the number of annual layers counted to a certain depth where a time stratigraphic marker, such as a known volcanic eruption or the radioactivity signal of nuclear debris) is present to the number of expected years established with the time marker; however, this comparison only indicates the accuracy of counting, not precision. In practice, the counting is often revised by reclassifying

ambiguous layers, if its result differs significantly from the expected number of years, to reconcile with the age of the layer where the time marker is, in order to improve accuracy. To address precision directly, researchers have tried several approaches to get a quantitative sense of ALC precision/uncertainty. For example, Alley et al. (1997) estimated ALC uncertainty based on the difference of the number of layers counted by different individuals and/or at different times by the same individual. Rasmussen et al. (2006) and Ferris et al. (2012) derived ALC uncertainty by summing up the number of ambiguous (possible and possibly not) annual layers.

This paper by Zhang et al. does not discuss the uncertainty in the identification of the annual layers using elemental data (concentrations ought to be proportional to counts per second in mass spectrometry). Usually, an annual cycle is defined as one maximum and one minimum of the measured concentration of the chemical species (in some instances a measured physical property may be used), at least this is the case for polar ice cores. For cores analyzed with high temporal resolution, cases of ambiguous annual layers are rare, leading to small uncertainties. Because the deposition processes of chemical impurities on non-polar mountain glaciers are probably different (Lines 188-189) from those common in the polar regions, the authors use a definition of annual signal different from that often used on polar ice cores. In this work, the authors define (Lines 187-188) annual layers as groups of peaks (of element concentration) "separated by a prolonged section of low element concentrations". One reason for this is that multiple peaks of a given element are found in each annual layer (Lines 188-190). This definition of an annual layer may work in some cases or cores, but raises questions on what an annual layer is, such as how many peaks constitute "a group of peaks" and how long a section is considered "prolonged". The data presented in Figure S7 look very obvious that two years of accumulation are in about 4 cm of the core. But I don't see many years in Figure 1 which are so obvious. Additionally, how does one decide where the annual layer starts where it ends, when multiple chemical species do not agree with each other? An example of ambiguous or inconsistent layers in different species can be seen in the depth interval of 108.09-108.15 m in Section II (Figure 1). With this type of data and the definition of annual layers, the determination of annual layers and thickness is quite subjective, at least more so than in polar ice cores where the definition is clear and annual signals are often unambiguous.

The authors acknowledge the limitation, resulting from this annual layer definition, on the precision of layer determination in Line 192: "identification of annual layers requires expert judgment". To me, this means that researchers with varying degree of experience or "expertise" or different perspectives will count differently. The results will be different, not only in the total number of annual layers counted (accuracy), but also in the thickness of individual layers. This is critically important to the objective of this work – accumulation history – as uncertainty in layer thickness leads directly to uncertainty in accumulation of individual years as well as long-term accumulation trends.

I would like to see some discussion, hopefully supported with data, on the uncertainty of the annual layer thickness determination in the core(s) studied in this work and using the counting method. Also, it would be helpful to explain what the expertise is to make "expert judgment", so as to provide a measure of the objectivity of layer counting.

Response: We agree with the reviewer that the accuracy and uncertainty of annual layer counting (ALC) is critically important to the success of layer thickness determination and ice core dating. ALC is always accompanied by a certain level of subjectivity. Such as the example given by the reviewer, Alley et al. (1997) estimated ALC uncertainty based on the difference of the number of layers counted by different individuals and/or at different times by the same individual. This could be considered "expert judgment". This is also why we included the results of annual layer identification from the StratiCounter algorithm to provide a relatively "objective" mathematic expression in comparison with our subjective visual identifications.

In the revision, we follow the approach successfully employed for Greenland ice cores by Rasmussen et al. (2006) to quantify counting uncertainty from uncertain layers. In this approach, we count uncertain layers as 0.5 ±

0.5 years, and estimate the maximum counting error (MCE) from the number of uncertain layers (N) as N × 0.5 years. Using the records for four elements (i.e., Al, Ca, Fe, Mg), we define "uncertain annual layer boundaries" as those without synchronous peaks of all four elements. Annual layer peaks for each of the three sections and their respective uncertainties are shown in Fig. 1. The number of annual layers for Section I, II, and III is 8 ± 2, 19 ± 3, and 23 ± 3. The derived average annual layer thickness for Section I, II, and III is thus $38.30 \pm_{6.38}^{9.57}$ mm (corresponding to $30.96 \pm_{5.16}^{7.74}$ mm w.e.), $18.42 \pm_{2.51}^{3.45}$ mm ($14.74 \pm_{2.01}^{2.76}$ mm w.e.), and $12.71\pm_{1.47}^{1.91}$ mm ($10.16 \pm_{1.17}^{1.52}$ mm w.e.), respectively. We included these updated results in the revision.

[Figure]

Figure 1. Annual layer counting for the Top Section, Section I, II, and III (top to bottom). The annual layers of the Top Section are identified based on the seasonality of $\delta^{18}$O and two $\beta$-activity peaks (An et al., 2016). The annual layers of Section I, II, and III are marked at the winter/spring peaks (grey bars) of Al, Ca, Fe, and Mg concentrations. The open grey bars filled with forward slashes indicate uncertain annual layers. Thin grey lines indicate raw data, and thick colored lines represent 200-point Gaussian smoothing. LA-ICP-MS intensity is reported as counts per

second.

I am curious why the authors decided to present data and offer discussion on only three very short sections of Core 2 of approximately 135 meters. It appears that the chemical analysis and layer counting were done for most, if not all, of the core: the number of years counted extends to the deepest of the core (Figure 1).

Response: So far, we have only performed the LA-ICP-MS measurement on the three trial sections. We plan to perform more LA-ICP-MS measurements in the near future.

References

Alley, R. B., et al. (1997), Visual-stratigraphic dating of the GISP2 ice core: Basis, reproducibility and application, J. Geophys. Res., 102(C12), 26,367– 26,381.

Ferris, D. G., J. Cole-Dai, A. R. Reyes, and D. M. Budner (2011), South Pole ice core record of explosive volcanic eruptions in the first and second millennia AD and evidence of a large eruption in the tropics around 535 AD, J Geophys Res-Atmos, 116, doi:10.1029/2011JD015916.

Rasmussen, S. O., et al. (2006), A new Greenland ice core chronology for the last glacial termination, J. Geophys. Res., 111, D06102, doi:10.1029/2005JD006079.

Sigg, A. and A. Neftel, (1980) Seasonal variations of hydrogen peroxide in polar ice core, Ann. Galciol., 10, 157-162.

Sigg, A., K. Fuher, M. Anklin, T. Staffelbach, and D. Zurmuhle (1994), A continuous analysis technique for trace species in ice cores, Env. Sci. & Tech., 28, 204-206.

Sigl, M., et al. (2016), The WAIS Divide deep ice core WD2014 chronology - Part 2: Annual-layer counting (0–31 ka BP), Climate of the Past, 12, 769-786, doi:10.5194/cp-12-769-201.

**References**

Alley, R. B., Shuman, C. A., Meese, D. A., Gow, A. J., Taylor, K. C., Cuffey, K. M., Fitzpatrick, J. J., Grootes, P. M., Zielinski, G. A., Ram, M., Spinelli, G., and Elder, B.: Visual-stratigraphic dating of the GISP2 ice core: Basis, reproducibility, and application, J. Geophys. Res.-Oceans, 102, 26367–26381, https://doi.org/10.1029/96jc03837, 1997.

An, W., Hou, S., Zhang, W., Wu, S., Xu, H., Pang, H., Wang, Y., and Liu, Y.:. Possible recent warming hiatus on the northwestern Tibetan Plateau derived from ice core records, Sci. Rep., 6, 32813, https://doi.org/10.1038/srep32813, 2016.

Hou, S., Zhang, W., Fang, L., Jenk, T. M., Wu, S., Pang, H., and Schwikowski, M.: Brief communication: New evidence further constraining Tibetan ice core chronologies to the Holocene, The Cryosphere, 15, 2109–2114, https://doi.org/10.5194/tc-15-2109-2021, 2021.

Hou, S., Zhang, W., Pang, H., Wu, S.-Y., Jenk, T. M., Schwikowski, M., and Wang, Y.: Apparent discrepancy of Tibetan ice core δ18O records may be attributed to misinterpretation of chronology, The Cryosphere, 13, 1743–1752, https://doi.org/10.5194/tc-13-1743-2019, 2019.

Hou, S., Jenk, T. M., Zhang, W., Wang, C., Wu, S., Wang, Y., Pang, H., and Schwikowski, M.: Age ranges of the Tibetan ice cores with emphasis on the Chongce ice cores, western Kunlun Mountains, The Cryosphere, 12, 2341–2348, https://doi.org/10.5194/tc-12-2341-2018, 2018.

Rasmussen, S. O., Andersen, K. K., Svensson, A. M., Steffensen, J. P., Vinther, B. M., Clausen, H. B., Siggaard-Andersen, M. L., Johnsen, S. J., Larsen, L. B., Dahl-Jensen, D., Bigler, M., Röthlisberger, R., Fischer, H., Goto-Azuma, K., Hansson, M. E., and Ruth, U.: A new Greenland ice core chronology for the last glacial termination, J. Geophys. Res.-Atmos., 111, 1–16, https://doi.org/10.1029/2005JD006079, 2006.

---

## Referee Report (RR1)

Comments to specific items of text are refered as Pxxx Lyyy for Page xxx Line yyy

**1 General Comments**

Zhang et al. "A new method of resolving annual precipitation for the past millennia from Tibetan ice cores" presents a detailed study on the average accumulation rate for 3 epochs in the last 2500 years for an ice core site on the Chongce ice cap, northwestern Tibetan Plateau. The paper combines annual layer thickness data (from ultra-high resolution ice core elemental chemistry) with a flow thinning model (constrained by water-insoluble organic carbon $^{14}$C ages) to determine local net accumulation over 3 disjoint epochs. The authors have done a commendable job of addressing the issues raised during the previous review. In particular, I appreciate the information and revisions around my previous major comment concerning the data fitting for the flow model. However, there are some inconsistencies between the author response to the reviews (version 3-1) and the manuscript (version 5-2), at least for the versions I accessed. I recommend minor alterations and corrections detailed below.

**2 Inconsistencies between authors response to reviews and manuscript**

**2.1 Minor Specific Comments**

Original comment:

P2 2nd paragraph. This needs a restructure, at the moment, the sentence topics are annual layers, thinning, annual layers then thinning again. Suggest you move the sentence staring "In addition, the nonlinear" to after the sentence staring "The most common approach". Then change "The thinning parameter" → "This thinning parameter".

The authors response is:

```
We agree with the reviewer, and have revised the sentence accordingly.
The revised sentence is as follows;
The most common approach is to obtain annual-layer thickness based
on the seasonal cycles of ice core parameters such as stable isotope
ratio of oxygen in the water (d18O), the concentration of major ions
(e.g. Ca2+, Mg2+, NH+4 , SO2-4), and the presence of melt layers
(Thompson et al., 2018). In addition, the nonlinear thinning of annual
layers caused by ice flow must be suitably constrained (Bolzan, 1985;
Henderson et al., 2016; Roberts et al., 2015).
```

This change is not reflected in the revised manuscript, with the second sentence still being before the first (as it was in the original manuscript).

Original comment:

P6 L188–189 These grouped peaks could also be from independent snow events with dry wind blown dust deposition between these snow events.

The authors response is:

```
We agree with the reviewer, and have revised the text accordingly.
The revised sentence is as follows;
These grouped peaks are interpreted as independent snow events
with elevated element concentrations or with wind-blown dust
deposition between these snow events.
```

There is no such sentence in the revised manuscript.

**2.2 Technical corrections**

P6 L149 Delete "were". Still need to delete first occurrence of "were", now on line 157, i.e. change "in the ice were by filtration" to "in the ice by filtration"

There were several other cases where the authors response to minor comments stated that "Change has been made accordingly" but the manuscript does not reflect this. However, the cases not listed here were only for suggested word changes rather than an error, so I have not listed them.

**3 New comments**

**3.1 Technical corrections**

P3 L64, I think neither Roberts et al 2015 or Winstrup et al 2012 are appropriate references for alpine ice cores. Suggest either find a more appropriate reference, or reword sentence to "because of the difficulties in identifying annual layers and obtaining accurate chronologies in the deeper part of ice cores due to rapid thinning (Roberts et al., 2015; Winstrup et al., 2012), with this issue especially problematic for alpine locations."

P3 L70, suggest adding "e.g.," to the start of the reference list.

P3 L83, change "Fig. S1" to "Fig. 1"

P3 L85–86, change "first time to preform the" to "first application of"

P7 L193, change "Fig.1" to "Fig. 4". If this reference is really figure 4 then need to swap the order of figures 3 and 4 so that they are first referenced in the correct order.

P7 L189, change "Fig.1" to "Fig. 2"

P9 L244, delete "roughly", the two age estimates agree within a small fraction of their uncertainties.

P11 L314–315, annual precipitation reconstructions based on ice cores has indeed already been achieved for both Greenland (e.g., Dahl-Jensen et al 1993) and Antarctica (e.g. Thomas et al 2017, doi:10.5194/cp-13-1491-2017). I suggest you include these (or other) references.

P12 L328, change "ice core model" to "ice cap flow model"

---

## Author Response (AR2)

Dear Dr. Carlos Martin,

Many thanks for your time and efforts in managing our contribution, and many thanks to the reviewers for their thoughtful comments. Below I have made a point-to-point response to all comments. The comments are in black, and my response is in blue. I have revised the original manuscript based on these comments. I hope the revised manuscript proves satisfactory.

Sincerely yours,

Hou Shugui
* * *
Editor Carlos Martin:

I want to thank again the reviewers for a thoughtful and detailed review, and the authors for the thoughtful and detailed response.

At this stage, after a major review, I always ask for another review round. I want to know the reviewer's opining to the revised manuscript. In particular here, a few fundamental concerns were raised: "the stress about the novelty of the method presented but, more importantly, the reviewers raise concerns about how data is fitted to the ice flow model and how the uncertainty in annual layer counting is propagated". I am going to activate another round of review now, but there is a point that I would like to raise before doing so.

Response: Many thanks for your thoughtful comments. To confirm the reliability of the dating results based on the ice flow model, we used Monte Carlo simulations (Breitenbach et al., 2012) to establish a continuous depth-age relationship of the Chongce Core 2 based on $^{14}$C ages and the $\beta$ activity horizon (An et al., 2016, Hou et al., 2018) (Fig. 1). This method was used before to establish the depth-age scale of the Mt. Ortles ice core extracted from the summit of Alto dell'Ortles in the Italian Alps (Gabrielli et al., 2016) and the Zangser Kangri ice core extracted from the northwestern Tibetan Plateau (Hou et al., 2021). This method can account for potential changes in snow accumulation and/or strain rate. The Monte Carlo-based annual layer thickness for the Top Section, Section I, II, and III are 122.04 mm w.e., 35.14 mm w.e., 40.79 mm w.e., and 9.06 mm w.e., respectively. For the Top Section, both the ice flow model and Monte Carlo simulations produced lower mean annual layer thickness than observed (140.76 mm w.e.). For Section I and II, both methods generated higher mean annual layer thicknesses than observed ($30.96 \pm _{5.16}^{7.74}$ mm w.e. and $14.74 \pm _{2.01}^{2.76}$ mm w.e.). For Section III, they produce comparable mean annual layer thicknesses to the observed value ($10.16 \pm _{1.17}^{1.52}$ mm w.e.). Therefore, both methods produced relatively consistent temporal patterns, despite some difference in the results.

The depth-age relationship from the Monte Carlo simulations was included in the *Supplement* as Figure S8.

[Figure]

**Figure 1.** The age-depth relationship for the Chongce Core 2 based on 2000 Monte Carlo simulations fitting the absolute dated age horizons. Solid black lines indicate the mean values, and dotted lines indicate the $1\sigma$ confidence interval. The red cross stands for the reference layer of the $\beta$ activity peak in 1963 A.D. (An et al., 2016). Blue circle show the individual calibrated WIOC [14]C ages, and the magenta dots represent the average of the CC-5 and CC-6, and the average of CC-7 and CC-8 ages at their average depths. Error bars represents the $1\sigma$ uncertainty. Note that the two magenta dots are included in the Monte Carlo simulation instead of the four original data points because the slight age reversals, but all the CC-5, CC-6, CC-7, and CC-8 are located within $1\sigma$ uncertainty range. In addition, CC-3 (non-tractable reversal) is regarded as an outlier and not included in the simulations, because its lower $2\sigma$ margin falls outside the upper $2\sigma$ margin of the subsequent point in the dating table.

We have also develop a method to quantify the propagation of uncertainty. Here we take Section I as an example. The length of Section I is 309.64 mm w.e. and its number of annual layer is 10 ± 2. Therefore, the mean annual layer thickness for Section I is 30.96 mm w.e. (i.e., 309.64 mm w.e. /10). If we take 12 and 8 years as the upper and lower limits for dating, the subsequent minimum and maximum annual layer thickness would be 25.80 mm w.e. (i.e., 309.64 mm w.e. /12) and 38.71 mm w.e. (i.e., 309.64 mm w.e. /8), respectively. For Section I, the observed upper limit of annual layer thickness (38.71 mm w.e.) was lower than that derived by the ice flow model (47.16 mm w.e.). It is also lower than the Holocene mean annual thickness, which supports the conclusion of this paper.

I wrote in my access review that I found the 'we present a new method' out of place. Reviewer #2 agreed with me and raised this point also. In the response you explain how it is "the first time that the ultra-high-resolution LA-ICP-MS measurement is applied to mountain glacier ice cores", and also " the first time that such annual layer identification is combined with layer thinning modelling to reconstruct annual precipitation records at the millennial time scale from the Tibetan ice cores". All

that is great and, as I said, justifies publication in TC, but the point is that those methods, LA-ICP-MS or layer thinning, are not new. However, the revised abstract states "In this study, we propose a new method to quantify annual accumulation". That is not true, and I can anticipate that I won't let the manuscript to be published as it is.

Response: We revised all references of "a new method" in the title, abstract and the text to "a quantitative method".

References

An, W., Hou, S., Zhang, W., Wu, S., Xu, H., Pang, H., Wang, Y., and Liu, Y.:. Possible recent warming hiatus on the northwestern Tibetan Plateau derived from ice core records, Sci. Rep., 6, 32813, https://doi.org/10.1038/srep32813, 2016.

Breitenbach, S. F. M., Rehfeld, K., Goswami, B., Baldini, J. U. L., Ridley, H. E., Kennett, D. J., Prufer, K. M., Aquino, V. V., Asmerom, Y., Polyak, V. J., Cheng, H., Kurths, J., and Marwan, N.: COnstructing Proxy Records from Age models (COPRA), Clim. Past, 8, 1765–1779, https://doi.org/10.5194/cp-8-1765-2012, 2012.

Gabrielli, P., Barbante, C., Bertagna, G., Bertó, M., Binder, D., Carton, A., Carturan, L., Cazorzi, F., Cozzi, G., Dalla Fontana, G., Davis, M., De Blasi, F., Dinale, R., Dragà, G., Dreossi, G., Festi, D., Frezzotti, M., Gabrieli, J., Galos, S. P., Ginot, P., Heidenwolf, P., Jenk, T. M., Kehrwald, N., Kenny, D., Magand, O., Mair, V., Mikhalenko, V., Lin, P. N., Oeggl, K., Piffer, G., Rinaldi, M., Schotterer, U., Schwikowski, M., Seppi, R., Spolaor, A., Stenni, B., Tonidandel, D., Uglietti, C., Zagorodnov, V., Zanoner, T., and Zennaro, P.: Age of the Mt. Ortles ice cores, the Tyrolean Iceman and glaciation of the highest summit of South Tyrol since the Northern Hemisphere Climatic Optimum, The Cryosphere, 10, 2779–2797, https://doi.org/10.5194/tc-10-2779-2016, 2016.

Hou, S., Zhang, W., Fang, L., Jenk, T. M., Wu, S., Pang, H., and Schwikowski, M.: Brief communication: New evidence further constraining Tibetan ice core chronologies to the Holocene, The Cryosphere, 15, 2109–2114, https://doi.org/10.5194/tc-15-2109-2021, 2021.

Anonymous Referee #1:

**1 General Comments**

Zhang et al. "A new method of resolving annual precipitation for the past millennia from Tibetan ice cores" presents a detailed study on the average accumulation rate for 3 epochs in the last 2500 years for an ice core site on the Chongce ice cap, northwestern Tibetan Plateau. The paper combines annual layer thickness data (from ultra-high resolution ice core elemental chemistry) with a flow thinning model (constrained by water-insoluble organic carbon $^{14}$C ages) to determine local net accumulation over 3 disjoint epochs. The paper is well written and structured and generally presents sufficient supporting evidence. I recommend minor alterations and corrections detailed below.

**2 Specific Comments**

**2.1 Major specific comment**

The most significant problem with the manuscript as it stands is the data fit to the flow model presented in Figure 2 and associated text on P8 L228-229. In particular, it appears that all of the $^{14}$C ages are above the nonlinear least squares data fit. This raises questions about the validity of the data

fit and if the solution has converged. I would have expected at least some of the $^{14}$C ages to be below the data fit. Specially, the data fit line can be moved upward and this would reduce the error at every observational data point, and hence the overall error of the fit. The authors need to verify that the data fit presented is indeed a (near) optimal fit, and redo the accumulation analysis if the data fit needs to be revised and improved.

Response: In this study, the depth-age relationship of the Chongce 135.81 m Core 2 was established by using a two-parameter (2p) model. The 2p model was first constrained by the $^{14}$C calibrated ages, together with the $\beta$-activity reference horizon of the Chongce 58.82 m Core 3, located only ~ 2 meters apart (Hou et al., 2018; Pang et al., 2020). We found that by using these data only, the 2p model is poorly constrained at the deep section, and giving an estimate bottom age much older than the bottom age ($8.3 \pm _{3.6}^{6.2}$ ka B.P.) estimated for Core 4 (Hou et al., 2018). Therefore, we included the Core 4 bottom age to constrain the final 2p model. Due to its mathematical configuration to account for ice flow dynamics, the 2p model gives more weight to points at shallower sections. Therefore, the inclusion of the Core 4 bottom age (relatively younger than otherwise derived bottom age) pushes the curve towards the left (younger) of most $^{14}$C dates. However, we believe this model gives the most reasonable results, compared with several other model fit based on different data combinations (Figure 1). The details of these model fits are provided as follows.

(1) all data (including $\beta$-activity peak of Core 3 and nine $^{14}$C ages) (Fig. 1a).

Results: The derived annual accumulation rate of $137 \pm 54$ mm w.e./year is in good agreement with the value of 140 mm w.e./year based on the tritium horizon. But the model is poorly constrained in deeper sections: the derived age estimate at the depth of the deepest $^{14}$C sample is $9.1 \pm _{4.0}^{7.2}$ ka B.P.. This is much older than the actual measured $^{14}$C age of $6.3 \pm 0.2$ ka B.P. at that depth (Fig. 1a).

(2) all data (including $\beta$-activity peak of Core 3 and nine $^{14}$C ages) and constant accumulation rate (140 m w.e./year) (Fig. 1b).

Results: The derived ice age at the bedrock is $30.7 \pm _{18.4}^{44.8}$ ka B.P., which is much older than the bottom age ($8.3 \pm _{3.6}^{6.2}$ ka B.P.) estimated for Core 4. In addition, the derived age estimate at the depth of the deepest $^{14}$C sample is $9.2 \pm _{3.6}^{6.0}$ ka B.P.. This is much older than the $^{14}$C age of $6.3 \pm 0.2$ ka B.P. at that depth. (Fig. 1b).

(3) $\beta$-activity peak of Core 3 and oldest six $^{14}$C ages (Fig. 1c).

Results: The derived ice age at the bedrock is $22.5 \pm _{13.8}^{34.8}$ ka B.P., which is much older than the bottom age ($8.3 \pm _{3.6}^{6.2}$ ka B.P.) estimated for Core 4. In addition, the derived accumulation ($233 \pm 104$ mm w.e./year) deviates significantly from the $\beta$-activity based estimate (140 mm w.e./year) (Fig. 1c).

(4) $\beta$-activity peak of Core 3, oldest six $^{14}$C ages, and constant accumulation rate (140 mm w.e./year) (Fig. 1d).

Results: The derived ice age at the bedrock is $50.1 \pm _{35.6}^{118.4}$ ka B.P., which is much older than the bottom age ($8.3 \pm _{3.6}^{6.2}$ ka B.P.) estimated for Core 4. In addition, the derived age estimate at the depth of the deepest $^{14}$C sample is $9.6 \pm _{4.1}^{7.3}$ ka B.P.. This is much older than the $^{14}$C age of $6.3 \pm 0.2$ ka B.P. at that depth (Fig. 1d).

(5) all data (including $\beta$-activity peak of Core 3 and nine $^{14}$C ages) plus bedrock estimate from Core 4 (Hou et al., 2018) as an additional model input point (**the method used in this manuscript**) (Fig. 1e).

Results: The derived ice age at the bedrock is $9.0 \pm _{3.6}^{7.9}$ ka B.P., which is roughly consistent with the bottom age ($8.3 \pm _{3.6}^{6.2}$ ka B.P.) estimated for Core 4. The derived accumulation rate ($103 \pm 34$ mm w.e./year) is in relative agreement with the $\beta$-activity based estimate (140 mm w.e./year). In addition,

the modeled age at the depth of the deepest $^{14}$C sample is now $5.2 \pm {}^{1.9}_{1.2}$ ka B.P. which, with the uncertainty range, is similar to the $^{14}$C age of $6.3 \pm 0.2$ ka B.P. (Fig. 1e). We believe this model provides most reasonable results, and is therefore adopted for this paper.

[Figure]

Figure 1. The depth-age relationship of the Chongce Core 2 based on the two-parameter model.

**2.2 Minor specific comment**

P2 L40 Is Christiansen and Ljungqvist (2017) the correct citation? This paper is about temperature reconstruction, and only mentions precipitation because of its influence on temperature reconstructions. Response: We replaced this citation with Sun et al., 2018, which presented a comprehensive review of the data sources and estimation methods of 30 currently available global precipitation data sets, including gauge-based, satellite-related, and reanalysis data sets.

P2 2$^{nd}$ paragraph. This needs a restructure, at the moment, the sentence topics are annual layers, thinning, annual layers then thinning again. Suggest you move the sentence starting "In addition, the nonlinear" to after the sentence starting "The most common approach". Then change "The thinning

parameter" → "This thinning parameter".

Response: We agree with the reviewer, and have revised the sentence accordingly. The revised sentence is as follows;
*The most common approach is to obtain annual-layer thickness based on the seasonal cycles of ice core parameters such as stable isotope ratio of oxygen in the water ($\delta^{18}O$), the concentration of major ions (e.g. $Ca^{2+}$, $Mg^{2+}$, $NH_4^+$, $SO_4^{2-}$), and the presence of melt layers (Thompson et al., 2018). In addition, the nonlinear thinning of annual layers caused by ice flow must be suitably constrained (Bolzan, 1985; Henderson et al., 2006; Roberts et al., 2015).*

P3 L80 I think the location map (Fig. S1) should be moved into the main manuscript, as this is key information.
Response: We agree with the reviewer, and have included the location map (Fig. S1) in the main text of the manuscript.

P5 Section 2.3 You do not give the vertical size of the samples required to give the 1kg sample, this is key information for the depth uncertainty estimate of the $\beta$-activity dating.
Response: We have given details on ice samples for $\beta$-activity measurements (Table S1) in the supporting information.

Table S1. Details on ice samples for $\beta$-activity measurements.

| Sample # | Depth (m) | Depth (m w.e.) | Length (m) | Length (m w.e.) | $\beta$ activity (dph kg$^{-1}$) |
|---|---|---|---|---|---|
| 1 | 0.000-0.710 | 0.000-0.406 | 0.710 | 0.406 | 555.1 |
| 2 | 0.710-1.150 | 0.406-0.771 | 0.440 | 0.365 | 936.5 |
| 3 | 1.150-1.720 | 0.771-1.253 | 0.570 | 0.482 | 597.9 |
| 4 | 1.720-2.185 | 1.253-1.648 | 0.465 | 0.395 | 499.2 |
| 5 | 2.185-2.575 | 1.648-1.981 | 0.390 | 0.333 | 505.6 |
| 6 | 2.575-2.945 | 1.981-2.297 | 0.370 | 0.316 | 539.1 |
| 7 | 2.945-3.355 | 2.297-2.648 | 0.410 | 0.351 | 416.7 |
| 8 | 3.355-3.890 | 2.648-3.110 | 0.535 | 0.462 | 518.4 |
| 9 | 3.890-4.350 | 3.110-3.504 | 0.460 | 0.393 | 396.1 |
| 10 | 4.350-4.805 | 3.504-3.889 | 0.455 | 0.385 | 439.4 |
| 11 | 4.805-5.270 | 3.889-4.288 | 0.465 | 0.399 | 1754.5 |
| 12 | 5.270-5.780 | 4.288-4.735 | 0.510 | 0.447 | 385.8 |
| 13 | 5.780-6.320 | 4.735-5.198 | 0.540 | 0.463 | 504.9 |
| 14 | 6.320-6.780 | 5.198-5.593 | 0.460 | 0.395 | 749.1 |
| 15 | 6.780-7.200 | 5.593-5.948 | 0.420 | 0.355 | 963.2 |
| 16 | 7.200-7.690 | 5.948-6.362 | 0.490 | 0.414 | 224.9 |
| 17 | 7.690-8.170 | 6.362-6.767 | 0.480 | 0.406 | 1709.9 |
| 18 | 8.170-8.630 | 6.767-7.158 | 0.460 | 0.390 | 1910.3 |
| 19 | 8.630-9.120 | 7.158-7.571 | 0.490 | 0.413 | 479.9 |
| 20 | 9.120-9.580 | 7.571-7.977 | 0.460 | 0.407 | 574.2 |
| 21 | 9.580-10.020 | 7.977-8.361 | 0.440 | 0.384 | 98.6 |

| 22 | 10.020-10.550 | 8.361-8.819 | 0.530 | 0.457 | 682.8 |
|----|---------------|-------------|-------|-------|-------|
| 23 | 10.550-11.060 | 8.819-9.254 | 0.510 | 0.435 | 262.6 |
| 24 | 11.060-11.490 | 9.254-9.618 | 0.430 | 0.364 | 503.8 |
| 25 | 11.490-12.015 | 9.618-10.061 | 0.525 | 0.444 | 705.8 |
| 26 | 12.015-12.525 | 10.061-10.494 | 0.510 | 0.433 | 168.7 |
| 27 | 12.525-12.925 | 10.494-10.833 | 0.400 | 0.339 | 282.9 |
| 28 | 12.925-13.375 | 10.833-11.203 | 0.450 | 0.370 | 191.8 |
| 29 | 13.375-13.845 | 11.203-11.608 | 0.470 | 0.405 | 673.8 |
| 30 | 13.845-14.305 | 11.608-11.999 | 0.460 | 0.392 | 269.3 |
| 31 | 14.305-14.805 | 11.999-12.410 | 0.500 | 0.411 | 324.3 |

P5 L130 Was the Argon gas flow purged or was the system purged using Argon gas? If the later, suggest changing "whilst the Argon (Ar) gas flow was purged for two minutes" → "whilst the system was purged with Argon (Ar) gas for two minutes".

Response: We thank the reviewer for clarification, and have revised this sentence accordingly, as "whilst the system was purged with Argon (Ar) gas for two minutes".

P5 Section 2.4 you do not give the vertical size of the samples used for the $^{14}$C extraction, this is key information for the uncertainty estimate of the $^{14}$C dating, as there is uncertainty in both the age and depth.

Response: We have given the vertical size of the samples used for the $^{14}$C extraction in the supporting information.

P6 L188-189 These grouped peaks could also be from independent snow events with dry wind blown dust deposition between these snow events.

Response: We agree with the reviewer, and have revised the text accordingly. The revised sentence is as follows;

*These grouped peaks are interpreted as independent snow events with elevated element concentrations or with wind-blown dust deposition between these snow events.*

P9 L241-242 Make it clear that you are using the values of "b" and "p" that you found in Section 3.2.

Response: We have revised this sentence as "where $L_{(z)}$ is the modeled annual layer thickness (mm w.e.) for the average accumulation rate (b, i.e., 103 ± 34 mm w.e.) at the depth of $z$ given the thinning parameter of $p$ (i.e., 0.008).".

P10 L257 Change "can be securely stored" → "is preserved". In fact your density profiles (Fig. S6) suggest this for Core 2 and 3, which both lack the lower densities near the surface indicative of snow. I suggest you add a sentence at Line 258 making this point.

Response: Following the reviewer's comment, we have revised this sentence as "However, not all snowfall is preserved in high-elevation glaciers, due to wind scouring, snow drifting, and sublimation (Hardy et al., 2003). Moreover, firnification process might develop rapidly as indicated from the lack

the lower density layers (indicative of snow) near the glacier surface (Fig. S6)".

P10 L265 You have presented all other accumulation rates as mm w.e./yr, suggest that you do the same for the Thompson et al (2006) results, to allow for easy comparison.
Response: We agree with the reviewer, but because the density profile of the Guliya ice core is not available (Thompson et al., 1995), we are not able to calculate the accumulation rate of the Guliya ice core as mm w.e./yr, but this comparison is still reasonable given the similar density for the periods of 1950-1989 A.D. and 1160-1169 A.D.

P10 L284-286 This statement is not correct. For example, an error in either the $^{14}$C dating, or the flow model fit (see main points above) will introduce an error in the flow thinning model, which due to its non-linear nature will result in different relative average accumulations over various epochs.
Response: We agree with the reviewer. For this reason, we deleted this statement in the revision.

P11 L295-299 In fact you already have 9 such markers from the $^{14}$C age ties, which allow you to calculate the average accumulation rate over the 8 epochs these 9 makers define.
Response: This suggestion is theoretically possible, but we are not able to calculate the average accumulation rates over the 8 epochs between the 9 $^{14}$C age ties because the errors of the $^{14}$C ages cause overlaps of some ages.

Supp info, Figure 1b give details of where the remote sensing data is from, what is the instrument (e.g. optical, SAR) and give a data citation.
Response: We have included details about the remote sensing data, and provided a citation in the supporting information.

Supp info, Figure S8 Give details of which core (or cores) are being compared here.
Response: We have included details of the ice cores in the supporting information.

Supp info, Table S1 is the depth in meters water equivalent? Explain the difference between "$^{14}$C age" and "cal age".
Response: The depth in Table S2 is the measured depth in the field. For convenience of the readers, we also included the depth in meters water equivalent in the revision after taking account of the density profile.

Regarding "$^{14}$C age" and "cal age", "$^{14}$C age" denotes conventional radiocarbon age, which is calculated from the formula below:

$$t = -8033 \times \ln (Fs)$$

where $t$ is conventional radiocarbon age, Fs is the $^{14}$C / $^{12}$C ratio of the sample divided by the same ratio of the modern standard. "cal age" denotes the calibrated age using OxCal v4.3 (Ramsey and Lee, 2013) with the Northern (IntCal13) calibration curve.

Table S2. Results of radiocabon measurements for the Chongce 135.81 m Core 2 ice core samples. For the calibrated calender year, ranges are given with 68.2% probality.

| Sample # | Depth (m) | Depth (m w.e.) | Mass (g) | WIOC (µg) | F$^{14}$C | $^{14}$C age (ka B.P.) | Calibrated age (ka B.P.) |
|---|---|---|---|---|---|---|---|
| CC-1 | 79.46-80.21 | 65.74-66.31 | 307.7 | 20.3 ± 1.2 | 0.81 ± 0.01 | 1.679 ± 0.078 | 1.445-1.704 |

| | | | | | | | |
|---|---|---|---|---|---|---|---|
| CC-2 | 88.82-89.56 | 73.31-73.92 | 302.9 | 24.3 ± 1.4 | 0.80 ± 0.01 | 1.831 ± 0.138 | 1.572-1.921 |
| CC-3 | 99.44-100.10 | 82.12-82.65 | 304.6 | 13.8 ± 0.9 | 0.68 ± 0.01 | 3.133 ± 0.161 | 3.157-3.560 |
| CC-4 | 110.58-111.35 | 91.48-92.10 | 342.6 | 24.9 ± 1.4 | 0.78 ± 0.01 | 2.037 ± 0.142 | 1.827-2.296 |
| CC-5 | 116.62-117.43 | 96.39-97.05 | 330.9 | 9.1 ± 0.7 | 0.69 ± 0.01 | 3.012 ± 0.164 | 2.978-3.377 |
| CC-6 | 122.64-123.36 | 101.40-101.98 | 338.6 | 17.6 ± 1.1 | 0.69 ± 0.01 | 2.944 ± 0.157 | 2.892-3.331 |
| CC-7 | 131.41-132.10 | 108.54-109.12 | 324.6 | 22.6 ± 1.3 | 0.59 ± 0.01 | 4.228 ± 0.176 | 4.451-5.036 |
| CC-8 | 132.65-133.51 | 109.59-110.31 | 392.7 | 23.6 ± 1.4 | 0.60 ± 0.01 | 4.169 ± 0.175 | 4.424-4.951 |
| CC-9 | 134.31-135.03 | 110.98-111.59 | 292.4 | 23.0 ± 1.4 | 0.51 ± 0.01 | 5.466 ± 0.201 | 5.997-6.443 |

**3 Technical corrections**

P2 L34 Kidd and Hoffman 2011 do not say "most important" only "variable parameter associated with atmospheric circulation". Delete "most important".
Response: Correction has been made accordingly.

P2 L45 "glacier" → "glaciers".

Response: Change has been made accordingly.

P2 L45-47 It is possible to obtain accumulation rates at time-scales other than annual from ice-cores. Suggest changing "obtain reliable annual-layer thickness information" → "obtain reliable layer thickness information for the relevant times-scales (typically annual, but may be centennial for low temporal resolution sites or studies)".
Response: Change has been made accordingly.

P2 L48 You are not constraining the thinning, you are compensating for it, suggest changing "constrained" → "compensated for".

Response: Change has been made accordingly.

P2 L57 There are many more ice core records than just the citations you list, suggest changing "(Alley" → "(e.g., Alley"

Response: Change has been made accordingly.

P3 L62 Change "methods. e.g., the" → "methods, for example the".

Response: Change has been made accordingly.

P3 L64 Remove the full stop after "technology".
Response: Change has been made accordingly.

P3 L65 Maybe change "reveal" to "resolve".
Response: Change has been made accordingly.

P3 L69 Change "parameters" → "parameterisations"

Response: Change has been made accordingly.

P3 L77 Delete the word "parameter".
Response: Change has been made accordingly.

P3 L78 Change "record" → "records"

Response: Change has been made accordingly.

P4 L93 See comment above about moving Fig S1into the main manuscript.
Response: Change has been made accordingly.

P4 L96 Is there a citation for the statement that the local climate is "largely controlled by the mid-tropospheric westerlies"?
Response: Yes, we have added a citation (i.e., Pang et al. (2020)).

P4 L100 Given you have listed the summer (28%) and winter/spring (59%) precipitation percentages, also include for autumn (13%) rather than leave the reader to calculate this. Suggest changing "lowest

amount of precipitation." →

"lowest amount (13%) of precipitation."
Response: Change has been made accordingly.

P5 L145 There is some ambiguity about what you are removing the 3mm outer layer from, and while the reader can work it out, it is much better to make it easier for the reader to understand. Therefore, suggest changing "decontaminated the $^{14}$C samples" → "decontaminated the ice for the $^{14}$C samples".

Response: Change has been made accordingly.

P5 L147 The more common term is laminar flow "hood" rather than "box".
Response: Change has been made accordingly.

P6 L149 Delete "were"
Response: Change has been made accordingly.

P6 L153 Change "found in the previous studies (Uglietti et al., 2016)." → "found in Uglietti et al.

(2016). ".
Response: Change has been made accordingly.

P6 S2.5 You talk about verifying your annual-layer identification using StratiCounter, but at this point

in the manuscript you haven't described how you did your annual-layer identification. As this description comes later, suggest changing "To verify our annual-layer identifications" → "To verify our annual-layer identifications (see Section 3.1)".
Response: Change has been made accordingly.

P6 L166 While CCSM3 might have been "state-of-the-art" when this research was conducted (2006), this is no longer the case, with CCSM3 being replaced by CCSM4 in 2010. Suggest deleting "state-of-the-art".
Response: Change has been made accordingly.

P8 L208-209 Until this point your references have been in alphabetic order, so suggest you swap order of Rapp 2012 and Nye 1963.
Response: Change has been made accordingly.

P8 L225 Change "overweigh" → "over emphasise".
Response: Change has been made accordingly.

P9 L236 Change "of the Holocene" → "over the Holocene".
Response: Change has been made accordingly.

P9 L245 Change "The initial" → "The estimated original (pre-thinning)".
Response: Change has been made accordingly.

P13 L344 Change "Bronk Ramsey, C.," → "Ramsey, C. B.,".

Response: Change has been made accordingly.

P13 L350 Delete second, repeated "for large-scale temperature".
Response: Change has been made accordingly.

P15 L412 I don't think Parrenin et al 2004 is cited anywhere in the manuscript.
Response: We have deleted this citation in the revision.

P16 L443 I don't think Tang et al 2015 is cited anywhere in the manuscript.
Response: We have deleted this citation in the revision.

P17 L475 Change "sine" → "since".

Response: Change has been made accordingly.

Supp info, Figure S4 Change "The seasonal precipitation" → "Monthly precipitation".

Response: Change has been made accordingly.

**References**

Bolzan, J. F.: Ice flow at the Dome C ice divide based on a deep temperature profile, J. Geophys. Res., 90(D5), 8111–8124, https://doi.org/10.1029/JD090iD05p08111, 1985.

Hardy, D. R., Vuille, M., and Bradley, R. S.: Variability of snow accumulation and isotopic composition on Nevado Sajama, Bolivia. J. Geophys. Res, 108(D22), https://doi.org/10.1029/2003JD003623, 2003.

Henderson, K., Laube, A., Gäggeler, H. W., Olivier, S., Papina, T., and Schwikowski, M.: Temporal variations of accumulation and temperature during the past two centuries from Belukha ice core, Siberian Altai, J. Geophys. Res., 111, D03104, https://doi.org/10.1029/2005JD005819, 2006.

Hou, S., Jenk, T. M., Zhang, W., Wang, C., Wu, S., Wang, Y., Pang, H., and Schwikowski, M.: Age ranges of the Tibetan ice cores with emphasis on the Chongce ice cores, western Kunlun Mountains, The Cryosphere, 12, 2341–2348, https://doi.org/10.5194/tc-12-2341-2018, 2018.

Ramsey, C. B., and Lee, S.: Recent and planned developments of the program Oxcal, Radiocarbon, 55, 720–730, 2013.

Roberts, J., Plummer, C., Vance, T., van Ommen, T., Moy, A., Poynter, S., Treverrow, A., Curran, M., and George, S.: A 2000-year annual record of snow accumulation rates for Law Dome, East Antarctica, Clim. Past, 11, 697–707, https://doi.org/10.5194/cp-11-697-2015, 2015.

Pang, H., Hou, S., Zhang, W., Wu, S., Jenk, T. M., Schwikowski, M., and Jouzel, J.: Temperature Trends in the Northwestern Tibetan Plateau Constrained by Ice Core Water Isotopes Over the Past 7,000 Years, J. Geophys. Res. Atmos., 125(19), e2020JD032560, 2020.

Sun, Q., Miao, C., Duan, Q., Ashouri, H., Sorooshian, S., and Hsu, K.-L.: A review of global precipitation data sets: Data sources, estimation, and intercomparisons, Rev. Geophys., 56, 79–107, https://doi.org/10.1002/2017RG000574, 2018.

Thompson, L., Mosley-Thompson, E., Brecher, H., Davis, M., León, B., Les, D., Lin, P.-N., Mashiotta, T., and Mountain, K.:Abrupt tropical climate change: Past and present, P. Natl. Acad. Sci. USA, 103(28), 10536-10543, https://doi.org/10.1073/pnas.0603900103, 2006.

Thompson, L. G., Mosley-Thompson, E., Davis, M. E., Lin, P. N., Dai, J., and Bolzan, J. F.: A 1000 year climate ice-core record from the Guliya ice cap, China: its relationship to global climate variability, Ann. Glaciol., 21, 175–181, https://doi.org/ 10.1017/S0260305500015780, 1995.

Jihong Cole-Dai (Referee):

The paper's main objective is to reconstruct precipitation records from ice cores. The steps to accomplish the objective (Linea 80-88) are (1) determining thickness of annual layers in ice cores, (2) modeling layer thinning caused by ice flow, and (3) combining the results of (1) and (2) to reconstruct past precipitation. The authors call this process a "new method" to develop annual precipitation records from ice cores.

I am not sure what is new in the items presented in the paper. For Step 1, the way to identify annual layers in this work is the use of measurement of chemical impurities in ice cores that show annual oscillations in amount or concentration. But this is not new. Numerous papers have documented the identification of annual layers in ice cores using various chemical species measured at reasonable or acceptable temporal resolution and the use of the identified annual layers to derive layer thickness and the number of annual layers in a core (dating). Perhaps the authors mean that the chemical analysis

technique (LA-ICP-MS) is new to ice cores. But a measurement technique (elemental measurement) for Ca different from the other techniques (e.g., ion chromatography) does not necessarily make the method to identify annual layers new. For example, when hydrogen peroxide was first measured in ice cores and its concentration was found to oscillate annually, $H_2O_2$ measurement for annual layer counting was not considered a new method of dating or annual layer determination (Sigg and Neftel, 1980 and Sigg et al., 1994). Maybe the authors believe the millimeter LA-ICP-MS analysis resolution is new; but measurements such as electric conductivity (ECM) and dielectric profiling (DEP), which have been used for quite a while, can also be of high-resolution (mm or sub-mm).

Response: Although ice core annual layer identification and layer thinning modeling are not new, they have not been used to reconstruct detailed (e.g. annual) precipitation time series beyond a few tens of years from Tibetan ice cores due to technical challenges and limitations. The new method presented in study has the potential to develop annual precipitation time series at millennial time scale from alpine ice cores, which has not been achieved so far from any previous studies.

We agree with Prof. Cole-Dai that numerous papers have documented the identification of annual layers in ice cores using various chemical species measured at reasonable or acceptable temporal resolution. However, due to rapid layer thinning, annual layer identification was only possible for the top sections (meters or tens meters) of the Tibetan ice cores from the conventional discrete measurements (e.g., water stable isotopes, major ions, dust) used in these studies, with a typical sampling resolution of centimeters. Although ECM and DEP can be of high-resolution (mm or sub-mm) and have been widely used for polar ice cores (Rasmussen et al., 2006), both techniques have not been used for the Tibetan ice cores to establish their chronology. This is largely because the high dust background of these ice cores causes a high noise/signal ratio that obscures the seasonality of their ECM and DEP profiles.

Our study is indeed the first attempt to use LA-ICP-MS on a Tibetan ice core. Given the ultra-high sampling resolution of LA-ICP-MS (153 $\mu$m per sample in our study), two orders of magnitude higher than the sampling resolution (centimeters) of the conventional chemical measurements, this technique allows us to identify annual cycles in deeper sections of the Chongce ice core (back to 2.5 ka B.P. in this study, in comparison to the usual tens years based on the conventional discrete measurements).

It is also worth pointing out that the $^{14}$C measurements were first applied to establish the chronology of Tibetan Chongce ice cores (Hou et al., 2018, 2019 and 2021). This provides the necessary information of layer thinning to develop the new method. We wish that this method will be used to reconstruct a continuous high-resolution precipitation record at millennial time scale when LA-ICP-MS is performed on the entire Tibetan ice cores.

The paper title suggests that the authors think this is the first time ("new") that annual layer identification using high-resolution chemical measurement is applied to mountain glacier ice cores. But that point is not apparent or explicit in the paper body.

Response: It is indeed the first time that the ultra-high-resolution LA-ICP-MS measurement is applied to mountain glacier ice cores, making it possible to identify annual layers at deeper sections of these cores. It is also the first time that such annual layer identification is combined with layer thinning modeling to reconstruct annual precipitation records at the millennial time scale from the Tibetan ice cores. This new method has potential to be applied to other alpine ice cores. We have clarified these points in the revision.

For Step 2, the modeling of ice flow including the method to quantify layer thinning rate is obviously not new. So, reconstruction of original layer thickness using measured layer thickness and modeled thinning curve should not be considered new, if the layer identification method is not new. In fact, reconstruction of accumulation records from measured annual layer thickness and modeled and experimentally determined layer thinning has been used frequently for polar ice cores, with records as long as tens of thousands of years. Yet, the authors seem to suggest (Lines 59-60: "challenging to develop annually resolved accumulation records covering longer (e.g. millennial) time periods") that this is a rare accomplishment. Again, this would sound reasonable when referring to mountain glacier ice cores (but not so for polar ice cores).

Response: We agree with the reviewer that the novelty of the method only applies to mountain glacier ice cores. Annual layer identification can be achieved for the Greenland summit ice cores for the past tens of thousands of years (Rasmussen et al., 2006), but this is not the case regarding the mountain glacier ice cores due to their relatively short length and rapid thinning. Our study is indeed the first to apply this method, i.e. taking account of both annual layer identification at the millennial time scale and annual layer thinning, for the Tibetan ice cores. It has also potential to be applied to other alpine ice cores. We have clarified these points in the revision.

The authors state (Lines 187-198) that identification of annual layers using concentration cycles of the selected elements was "verified" with counting by the StratiCounter program. But the StratiCounter algorithm is not designed to count layers accurately. It is meant to facilitate the counting process and make counting less subjective. Therefore, counting with StratiCount ought not to be used to verify manual counting. In fact, the algorithm is supposed to be used only when the annual signals are clear and consistent (i.e., highly unambiguous; Sigl et al., 2015). In this case, with many annual signals highly ambiguous (multiple peaks in one annual layer; see Figure 1), result of Straticounting cannot be viewed as verification of the result of manual counting.

Response: We agree with the reviewer's concern on the StratiCounter program. Indeed, in this study, we relied on visual identification and manual counting as our primary method, using StratiCounter only for comparison purposes. In our case, the results from manual counting and StratiCounter are highly consistent, reinforcing the validity of our results. We revised the wording in the manuscript to clarify this.

The precision (i.e., uncertainty) of annual layer counting (ALC) is critically important to the success of layer thickness determination and ice core dating. Often times, researchers compare the number of annual layers counted to a certain depth where a time stratigraphic marker, such as a known volcanic eruption or the radioactivity signal of nuclear debris) is present to the number of expected years established with the time marker; however, this comparison only indicates the accuracy of counting, not precision. In practice, the counting is often revised by reclassifying ambiguous layers, if its result differs significantly from the expected number of years, to reconcile with the age of the layer where the time marker is, in order to improve accuracy. To address precision directly, researchers have tried several approaches to get a quantitative sense of ALC precision/uncertainty. For example, Alley et al. (1997) estimated ALC uncertainty based on the difference of the number of layers counted by different individuals and/or at different times by the same individual. Rasmussen et al. (2006) and

Ferris et al. (2012) derived ALC uncertainty by summing up the number of ambiguous (possible and possibly not) annual layers.

This paper by Zhang et al. does not discuss the uncertainty in the identification of the annual layers using elemental data (concentrations ought to be proportional to counts per second in mass spectrometry). Usually, an annual cycle is defined as one maximum and one minimum of the measured concentration of the chemical species (in some instances a measured physical property may be used), at least this is the case for polar ice cores. For cores analyzed with high temporal resolution, cases of ambiguous annual layers are rare, leading to small uncertainties. Because the deposition processes of chemical impurities on non-polar mountain glaciers are probably different (Lines 188-189) from those common in the polar regions, the authors use a definition of annual signal different from that often used on polar ice cores. In this work, the authors define (Lines 187-188) annual layers as groups of peaks (of element concentration) "separated by a prolonged section of low element concentrations". One reason for this is that multiple peaks of a given element are found in each annual layer (Lines 188-190). This definition of an annual layer may work in some cases or cores, but raises questions on what an annual layer is, such as how many peaks constitute "a group of peaks" and how long a section is considered "prolonged". The data presented in Figure S7 look very obvious that two years of accumulation are in about 4 cm of the core. But I don't see many years in Figure 1 which are so obvious. Additionally, how does one decide where the annual layer starts where it ends, when multiple chemical species do not agree with each other? An example of ambiguous or inconsistent layers in different species can be seen in the depth interval of 108.09-108.15 m in Section II (Figure 1). With this type of data and the definition of annual layers, the determination of annual layers and thickness is quite subjective, at least more so than in polar ice cores where the definition is clear and annual signals are often unambiguous.

The authors acknowledge the limitation, resulting from this annual layer definition, on the precision of layer determination in Line 192: "identification of annual layers requires expert judgment". To me, this means that researchers with varying degree of experience or "expertise" or different perspectives will count differently. The results will be different, not only in the total number of annual layers counted (accuracy), but also in the thickness of individual layers. This is critically important to the objective of this work – accumulation history – as uncertainty in layer thickness leads directly to uncertainty in accumulation of individual years as well as long-term accumulation trends.

I would like to see some discussion, hopefully supported with data, on the uncertainty of the annual layer thickness determination in the core(s) studied in this work and using the counting method. Also, it would be helpful to explain what the expertise is to make "expert judgment", so as to provide a measure of the objectivity of layer counting.

Response: We agree with the reviewer that the accuracy and uncertainty of annual layer counting (ALC) is critically important to the success of layer thickness determination and ice core dating. ALC is always accompanied by a certain level of subjectivity. Such as the example given by the reviewer, Alley et al. (1997) estimated ALC uncertainty based on the difference of the number of layers counted by different individuals and/or at different times by the same individual. This could be considered "expert judgment". This is also why we included the results of annual layer identification from the StratiCounter algorithm to provide a relatively "objective" mathematic expression in comparison with our subjective visual identifications.

In the revision, we follow the approach successfully employed for Greenland ice cores by Rasmussen et al. (2006) to quantify counting uncertainty from uncertain layers. In this approach, we

count uncertain layers as $0.5 \pm 0.5$ years, and estimate the maximum counting error (MCE) from the number of uncertain layers (N) as $N \times 0.5$ years. Using the records for four elements (i.e., Al, Ca, Fe, Mg), we define "uncertain annual layer boundaries" as those without synchronous peaks of all four elements. Annual layer peaks for each of the three sections and their respective uncertainties are shown in Fig. 1. The number of annual layers for Section I, II, and III is $8 \pm 2$, $19 \pm 3$, and $23 \pm 3$. The derived average annual layer thickness for Section I, II, and III is thus $38.30 \pm {}^{9.57}_{6.38}$ mm (corresponding to $30.96 \pm {}^{7.74}_{5.16}$ mm w.e.), $18.42 \pm {}^{3.45}_{2.51}$ mm ($14.74 \pm {}^{2.76}_{2.01}$ mm w.e.), and $12.71 \pm {}^{1.91}_{1.47}$ mm ($10.16 \pm {}^{1.52}_{1.17}$ mm w.e.), respectively. We included these updated results in the revision.

[Figure]

Figure 1. Annual layer counting for the Top Section, Section I, II, and III (top to bottom). The annual layers of the Top Section are identified based on the seasonality of $\delta^{18}O$ and two $\beta$-activity peaks (An et al., 2016). The annual layers of Section I, II, and III are marked at the winter/spring peaks (grey bars) of Al, Ca, Fe, and Mg concentrations. The open grey bars filled with forward slashes indicate uncertain

annual layers. Thin grey lines indicate raw data, and thick colored lines represent 200-point Gaussian smoothing. LA-ICP-MS intensity is reported as counts per second.

I am curious why the authors decided to present data and offer discussion on only three very short sections of Core 2 of approximately 135 meters. It appears that the chemical analysis and layer counting were done for most, if not all, of the core: the number of years counted extends to the deepest of the core (Figure 1).

Response: So far, we have only performed the LA-ICP-MS measurement on the three trial sections. We plan to perform more LA-ICP-MS measurements in the near future.

References

Alley, R. B., et al. (1997), Visual-stratigraphic dating of the GISP2 ice core: Basis, reproducibility and application, J. Geophys. Res., 102(C12), 26,367– 26,381.

Ferris, D. G., J. Cole-Dai, A. R. Reyes, and D. M. Budner (2011), South Pole ice core record of explosive volcanic eruptions in the first and second millennia AD and evidence of a large eruption in the tropics around 535 AD, J Geophys Res-Atmos, 116, doi:10.1029/2011JD015916.

Rasmussen, S. O., et al. (2006), A new Greenland ice core chronology for the last glacial termination, J. Geophys. Res., 111, D06102, doi:10.1029/2005JD006079.

Sigg, A. and A. Neftel, (1980) Seasonal variations of hydrogen peroxide in polar ice core, Ann. Galciol., 10, 157-162.

Sigg, A., K. Fuher, M. Anklin, T. Staffelbach, and D. Zurmuhle (1994), A continuous analysis technique for trace species in ice cores, Env. Sci. & Tech., 28, 204-206.

Sigl, M., et al. (2016), The WAIS Divide deep ice core WD2014 chronology - Part 2: Annual-layer counting (0–31 ka BP), Climate of the Past, 12, 769-786, doi:10.5194/cp-12-769-201.

**References**

Alley, R. B., Shuman, C. A., Meese, D. A., Gow, A. J., Taylor, K. C., Cuffey, K. M., Fitzpatrick, J. J., Grootes, P. M., Zielinski, G. A., Ram, M., Spinelli, G., and Elder, B.: Visual-stratigraphic dating of the GISP2 ice core: Basis, reproducibility, and application, J. Geophys. Res.-Oceans, 102, 26367–26381, https://doi.org/10.1029/96jc03837, 1997.

An, W., Hou, S., Zhang, W., Wu, S., Xu, H., Pang, H., Wang, Y., and Liu, Y.:. Possible recent warming hiatus on the northwestern Tibetan Plateau derived from ice core records, Sci. Rep., 6, 32813, https://doi.org/10.1038/srep32813, 2016.

Hou, S., Zhang, W., Fang, L., Jenk, T. M., Wu, S., Pang, H., and Schwikowski, M.: Brief communication: New evidence further constraining Tibetan ice core chronologies to the Holocene, The Cryosphere, 15, 2109–2114, https://doi.org/10.5194/tc-15-2109-2021, 2021.

Hou, S., Zhang, W., Pang, H., Wu, S.-Y., Jenk, T. M., Schwikowski, M., and Wang, Y.: Apparent discrepancy of Tibetan ice core $\delta^{18}O$ records may be attributed to misinterpretation of chronology, The Cryosphere, 13, 1743–1752, https://doi.org/10.5194/tc-13-1743-2019, 2019.

Hou, S., Jenk, T. M., Zhang, W., Wang, C., Wu, S., Wang, Y., Pang, H., and Schwikowski, M.: Age ranges of the Tibetan ice cores with emphasis on the Chongce ice cores, western Kunlun Mountains, The Cryosphere, 12, 2341–2348, https://doi.org/10.5194/tc-12-2341-2018, 2018.

Rasmussen, S. O., Andersen, K. K., Svensson, A. M., Steffensen, J. P., Vinther, B. M., Clausen, H. B., Siggaard-Andersen, M. L., Johnsen, S. J., Larsen, L. B., Dahl-Jensen, D., Bigler, M.,

Röthlisberger, R., Fischer, H., Goto-Azuma, K., Hansson, M. E., and Ruth, U.: A new Greenland ice core chronology for the last glacial termination, J. Geophys. Res.-Atmos., 111, 1–16, https://doi.org/10.1029/2005JD006079, 2006.

---

## Author Response (AR3)

Dear Dr. Carlos Martin,

Many thanks for your time and efforts in managing our contribution, and many thanks to the reviewers for their thoughtful comments. Below we have made a point-to-point response to the comments. The comments are in black, and our response is in blue. We have also revised the original manuscript based on these comments. I hope that the revised manuscript proves satisfactory.

Sincerely yours,

Hou Shugui
* * *
Anonymous Referee #1:

**1 General Comments**

Zhang et al. "A new method of resolving annual precipitation for the past millennia from Tibetan ice cores" presents a detailed study on the average accumulation rate for 3 epochs in the last 2500 years for an ice core site on the Chongce ice cap, northwestern Tibetan Plateau. The paper combines annual layer thickness data (from ultra-high resolution ice core elemental chemistry) with a flow thinning model (constrained by water-insoluble organic carbon [14]C ages) to determine local net accumulation over 3 disjoint epochs. The authors have done a commendable job of addressing the issues raised during the previous review. In particular, I appreciate the information and revisions around my previous major comment concerning the data fitting for the flow model. However, there are some inconsistencies between the author response to the reviews (version 3-1) and the manuscript (version 5-2), at least for the versions I accessed. I recommend minor alterations and corrections detailed below.

**2 Inconsistencies between authors response to reviews and manuscript**

2.1 Minor Specific Comments

Original comment:

P2 2nd paragraph. This needs a restructure, at the moment, the sentence topics are annual layers, thinning, annual layers then thinning again. Suggest you move the sentence staring "In addition, the nonlinear" to after the sentence staring "The most common approach". Then change "The thinning parameter"→ "This thinning parameter".

The authors response is:

We agree with the reviewer, and have revised the sentence accordingly. The revised sentence is as follows;

The most common approach is to obtain annual-layer thickness based on the seasonal cycles of ice core parameters such as stable isotope ratio of oxygen in the water ($\delta^{18}O$), the concentration of major ions (e.g. $Ca^{2+}$, $Mg^{2+}$, $NH_4^+$, $SO_4^{2-}$), and the presence of melt layers (Thompson et al., 2018). In addition, the nonlinear thinning of annual layers caused by ice flow must be suitably constrained (Bolzan, 1985; Henderson et al., 2006; Roberts et al., 2015). This change is not reflected in the revised manuscript, with the second sentence still being before the first (as it was in the original manuscript).

Response: We apologize for this negligence. This change has been made accordingly in the current revision (lines 49-54).

Original comment:

P6 L188-189 These grouped peaks could also be from independent snow events with dry wind blown dust deposition between these snow events.

The authors response is:

We agree with the reviewer, and have revised the text accordingly.

The revised sentence is as follows;

These grouped peaks are interpreted as independent snow events with elevated element concentrations or with wind-blown dust deposition between these snow events. There is no such sentence in the revised manuscript.

Response: We apologize for this negligence. This change has been made accordingly in the current revision (lines 198-200).

2.2 Technical corrections

P6 L149 Delete "were". Still need to delete first occurrence of "were", now on line 157, i.e. change "in the ice were by filtration" to "in the ice by filtration"

There were several other cases where the authors response to minor comments stated that "Change has been made accordingly" but the manuscript does not reflect this. However, the cases not listed here were only for suggested word changes rather than an error, so I have not listed them.

Response: We apologize for this negligence.

All changes stated in the previous response have been reflected in the current revision.

**3 New comments**

3.1 Technical corrections

P3 L64, I think neither Roberts et al 2015 or Winstrup et al 2012 are appropriate references for alpine ice cores. Suggest either find a more appropriate reference, or reword sentence to "because of the difficulties in identifying annual layers and obtaining accurate chronologies in the deeper part of ice cores due to rapid thinning (Roberts et al., 2015; Winstrup et al., 2012), with this issue especially problematic for alpine locations."

Response: We agree with this comment, and have replaced with two other appropriate references (Henderson et al., 2006; Yao et al., 2008).

P3 L70, suggest adding "e.g.," to the start of the reference list.

Response: Change has been made accordingly.

P3 L83, change "Fig. S1" to "Fig. 1"

Response: Change has been made accordingly.

P3 L85-86, change "first time to preform the" to "first application of"

Response: Change has been made accordingly.

P7 L193, change "Fig.1" to "Fig. 4". If this reference is really figure 4 then need to swap the order of figures 3 and 4 so that they are first referenced in the correct order.

Response: The "Fig. 1" has been deleted in the current revision.

P7 L189, change "Fig.1" to "Fig. 2"
Response: Change has been made accordingly.

P9 L244, delete "roughly", the two age estimates agree within a small fraction of their uncertainties.
Response: Change has been made accordingly.

P11 L314-315, annual precipitation reconstructions based on ice cores has indeed already been achieved for both Greenland (e.g., Dahl-Jensen et al 1993) and Antarctica (e.g. Thomas et al 2017, doi:10.5194/cp-13-1491-2017). I suggest you include these (or other) references.
Response: According to the comments from Dr. Cole-Dai (Referee), we have focused on the alpine ice cores, therefore, we deleted "Moreover, ice core accumulation records could be used to quantify annual precipitation over Antarctica and the Greenland ice cap, where no other precipitation proxies exist." in the current revision.

P12 L328, change "ice core model" to "ice cap flow model"
Response: Change has been made accordingly.

Jihong Cole-Dai (Referee):

I thank the authors for carefully and seriously responding to comments and questions from the reviewers and for addressing, in the revised manuscript, the major issues raised in those comments and questions.

Here I will comment on the response to my comment that the identification of annual layers and determination of annual accumulation (layer thickness and thinning correction) described in this manuscript is not new, as claimed in the original manuscript. The authors appear to agree with that comment and have removed the word "new" in presenting their method of annual accumulation determination. Their exact words are: "We agree with the reviewer that the novelty of the method only applies to mountain glacier ice cores. Annual layer identification can be achieved for the Greenland summit ice cores for the past tens of thousands of years (Rasmussen et al., 2006), but this is not the case regarding the mountain glacier ice cores due to their relatively short length and rapid thinning. Our study is indeed the first to apply this method, i.e. taking account of both annual layer identification at the millennial time scale and annual layer thinning, for the Tibetan ice cores. It has also potential to be applied to other alpine ice cores. We have clarified these points in the revision."

I will not dispute that this is the first time the approach of annual layer identification and modeling layer thinning is applied to an ice core from the Tibetan Plateau. But this paper is about METHODOLOGY of reconstructing accumulation/precipitation history (note that the paper does not present an accumulation record from the Chongce ice cores based on annual layer identification). Again, the method presented here does not qualify as new or novel.

I sense that what is new or what the authors truly would like to present as the major contribution with this work is the APPLICATION of that method (annual layer identification in conjunction with modeling layer thinning) to Tibetan Plateau ice cores and potentially to all mountain (alpine) glacier

ice cores. The "novelty", if novelty is necessary, is that the paper demonstrates that that method can work for mountain glacier ice cores, as it has worked well for polar ice cores. I feel the papers misses this point, perhaps due to the desire to present something "new" in (the entire field of) ice core research. I see this desire in numerous places in the paper, where the authors imply a new method (Lines 81-82) and stress a "first" (Lines 93-94). This tendency to be "first" leads to exaggerating statements. An example of that is "Moreover, ice core accumulation records could be used to quantify annual precipitation over Antarctica and the Greenland ice cap, where no other precipitation proxies exist." (Lines 313-315), where the authors seem to imply that their "proposed" method would be valuable beyond mountain glacier ice cores. There is just one problem with this implication: long (millennia) accumulation records have been constructed from Antarctica and Greenland ice cores based on the exact method (annual layer identification and layer thinning modeling).

I see that one flaw with the structure of the paper is that the description and presentation of approaches, methods and result discussion cover both polar ice cores and mountain glacier ice cores. When discussing shortcomings and limitations of previous work, the authors do not always distinguish between polar and non-polar ice cores. In Introduction, the authors, after describing how high-resolution measurements can discern annual layers in long polar ice cores covering thousands of years, state (Lines 71-73) "The remaining challenge for reconstructing long-term accumulation records thus lies in establishing accurate thinning parameters, and this is largely dependent on the reliable dating of ice cores, particularly at deeper sections." This statement may well apply to mountain glacier ice cores. But, in my opinion, this would be a very shaky statement about polar ice cores, at least for the millennial time scale.

Response: We agree with the directions, and have focused only on the alpine ice cores in this manuscript. Therefore, we deleted the contents concerning to the polar ice cores in the current revision. Specially, we have added "alpine" before "ice cores, particularly at deeper sections" in line 73 (line 73 in the current revision).

I would recommend that the paper be revised to make a clear distinction between polar ice cores and mountain/alpine glacier ice cores. I would like to see clear acknowledgement of (giving credit to) the success of reconstructing long accumulation records (using the method of annual layer identification and layer thinning modeling) with polar ice cores. This would allow the authors to claim and demonstrate "novel" approach or new application of existing methodology and true potential value of their work (to mountain glacier ice cores). Hopefully, such a revision would prevent exaggerations such as that in Lines 313-315.

Response: We agree with the reviewer. We have replaced "novel" with "quantitative" in the title and deleted "Moreover, ice core accumulation records could be used to quantify annual precipitation over Antarctica and the Greenland ice cap, where no other precipitation proxies exist." in the current revision.

Below are several comments and questions from me about a few specific aspects of the manuscript. I hope the authors will consider these when revising their manuscript.

Line 50. What are "low temporal resolution sites"?
Response: To prevent misunderstanding, we have deleted this sentence "but may be centennial for low temporal resolution sites" in line 50.

Line 50-51. How can non-linear thinning be "constrained"?

Response: We have adjusted the structure of the sentences in lines 49-58 . The revised sentences are as follows;

*The most common approach is to obtain annual-layer thickness based on the seasonal cycles of ice core parameters such as stable isotope ratio of oxygen in the water ($\delta^{18}O$), the concentration of major ions (e.g. $Ca^{2+}$, $Mg^{2+}$, $NH_4^+$, $SO_4^{2-}$), and the presence of melt layers (Thompson et al., 2018). In addition, the nonlinear thinning of annual layers caused by ice flow must be suitably compensated (Bolzan, 1985; Henderson et al., 2006; Roberts et al., 2015). This thinning parameter of ice cores is usually derived from an ice flow model constrained by the ages of absolute chronological markers, e.g., peak of beta and/or tritium activity from thermonuclear bomb testing in the second half of the $20^{th}$ century, well-defined aerosol layers and/or tephra from large volcanic eruptions, and radioactive dating method based on $^{210}Pb$ activity decay (Uglietti et al., 2016; Zhang et al., 2015).*

Lines 201-204. After describing how the uncertainty of the counted number of annual layers is determined, the uncertainty in reconstructed accumulation rate is presented. How is the accumulation uncertainty derived from the layer number uncertainty?

Response: The uncertainty of the mean annual layer thickness is derived from the layer number thickness. The upper limit of the mean annual layer thickness of each section was calculated through dividing the length of the section by the number of certain annual layers. The lower limit of the mean annual layer thickness of each section was calculated through dividing the length of the section by the total number of annual layers (including the number of certain and uncertain annual layers).

Line 205. In response to my previous comment, the authors "revised the wording" about using the StratiCounter program for layer counting. They removed the word "verify" regarding the program counting. However, here they state "the StratiCounter program (to) identify annual layers objectively". I wonder in what way the StratiCounter program is objective.

Response: The StratiCounter was designed to facilitate the counting process and make counting less subjective (Winstrup et al., 2012). In fact, the StratiCounter program only provide a relatively "objective" mathematic expression in comparison with subjective visual identifications. To avoid misunderstanding, we have deleted "objectively" in the revised manuscript.

Lines 304-305 (and 96-97). The authors claim that "the method proposed in this study produces reliable results and has the potential to reconstruct high-resolution continuous precipitation time series." The basis for the word "reliable" is entirely based on the comparison of the very limited results from this study with accumulation reconstructions from other studies of various methodology (Figure 4). The best word the authors use to characterize the comparison result is "consistent" (Lines 295 and 300). How does "consistent" lead to "verified" (Line 96) and "reliable"?

Response: We agree with the reviewer, and have changed "verified" to "evaluated" (line 96). We also revised the sentence in lines 304-305 as follows.

*These results suggest that the method proposed in this study has the potential to reconstruct high-resolution continuous precipitation time series.*